# LinkedSV for detection of mosaic structural variants from linked-read exome and genome sequencing data

Li Fang [1], Charlly Kao[2], Michael V. Gonzalez[2], Fernanda A. Mafra[2], Renata Pellegrino da Silva[2], Mingyao Li[3], Sören-Sebastian Wenzel[4], Katharina Wimmer [4], Hakon Hakonarson [5] & Kai Wang [1,6]*

Linked-read sequencing provides long-range information on short-read sequencing data by barcoding reads originating from the same DNA molecule, and can improve detection and breakpoint identification for structural variants (SVs). Here we present LinkedSV for SV detection on linked-read sequencing data. LinkedSV considers barcode overlapping and enriched fragment endpoints as signals to detect large SVs, while it leverages read depth, paired-end signals and local assembly to detect small SVs. Benchmarking studies demonstrate that LinkedSV outperforms existing tools, especially on exome data and on somatic SVs with low variant allele frequencies. We demonstrate clinical cases where LinkedSV identifies disease-causal SVs from linked-read exome sequencing data missed by conventional exome sequencing, and show examples where LinkedSV identifies SVs missed by high-coverage long-read sequencing. In summary, LinkedSV can detect SVs missed by conventional short-read and long-read sequencing approaches, and may resolve negative cases from clinical genome/exome sequencing studies.

[1] Raymond G. Perelman Center for Cellular and Molecular Therapeutics, Children's Hospital of Philadelphia, Philadelphia, PA 19104, USA. [2] Center for Applied Genomics, Children's Hospital of Philadelphia, Philadelphia, PA 19104, USA. [3] Department of Biostatistics, Epidemiology and Informatics, University of Pennsylvania, Philadelphia, PA 19104, USA. [4] Institute of Human Genetics, Department for Genetics and Pharmacology, Medical University of Innsbruck, Innsbruck, Austria. [5] Department of Pediatrics, University of Pennsylvania, Philadelphia, PA 19104, USA. [6] Department of Pathology and Laboratory Medicine, University of Pennsylvania, Philadelphia, PA 19104, USA. *email: wangk@email.chop.edu

Genomic structural variants (SVs) have been implicated in a variety of phenotypic diversity and human diseases[1]. Several approaches, such as split-reads[2,3], discordant read-pairs[3,4], and assembly based methods[5,6] have been developed for SV discovery from short reads. However, reliable detection of SVs from these approaches remains challenging. The split-reads and discordant read-pairs approaches require that the breakpoint-spanning reads/read-pairs are sequenced and confidently mapped. Genomic rearrangements are often mediated by repeats and thus breakpoint junctions of SVs are very likely to reside in repetitive regions[7–9]. Therefore, the breakpoint-spanning reads/read-pairs may be multi-mapped and have low mapping qualities. It is also difficult to perform assembly at repeat regions. Long-read sequencing, such as single-molecule real-time (SMRT) sequencing and nanopore sequencing are better for SV detection[10,11], but their application is limited by the higher cost and per-base error rate.

Linked-read sequencing technology developed by 10xGenomics combines the throughput and accuracy of short-read sequencing with the long-range information. In this approach, nanogram amounts of high-molecular weight (HMW) DNA molecules are dispersed into more than 1 million droplet partitions with different barcodes by a microfluidic system[12]. Thus, only a small number of HMW DNA molecules (~10) are loaded per partition[13]. The HMW DNA molecules can be up to several hundred kilobases in size and have a length-weighted mean DNA molecule length of about 50 kb. Within an individual droplet partition, HMW DNA molecules are primed and amplified by primers with a partition-specific barcode. The barcoded DNA molecules are released from the droplets and sequenced by standard Illumina paired-end sequencing[12]. The sequenced short reads derived from the same HMW DNA molecule can be linked together, providing long-range information for mapping, phasing and SV calling. In addition, linked-read whole-exome sequencing (WES) has also been developed[12], which provides an attractive and efficient option for clinical genetic testing.

In linked-read sequencing data, barcode similarities between any two nearby genome locations are very high, because the reads tend to originate from the same sets of HMW DNA molecules. In contrast, barcode similarities between any two distant genome locations are very low, because the reads of the two genome locations originate from two different sets of HMW DNA molecules and it is highly unlikely that two different sets of HMW DNA molecules share multiple barcodes. Thus, the presence of multiple shared barcodes between two distant locations indicates that the two distant locations are close to each other in the alternative genome[14]. A few pipelines and software tools have adopted this principle to call SVs from linked-read sequencing data, such as Longranger[12], GROC-SVs[14], and NAIBR[15]. Longranger is the official pipeline developed by 10x Genomics. Longranger bins the genome into 10 kb windows and finds the barcodes of high-mapping quality reads within each window. A binomial test is used to find all pairs of regions that are distant and share more barcodes than what would be expected by chance. A sophisticated probabilistic model is used to assign a likelihood and remove low-quality events[12]. GROC-SVs uses a similar method to find candidate SV loci but performed assembly to identify precise breakpoint locations. GROC-SVs also provides functionality to interpret complex SVs[14]. NAIBR detects SVs using a probabilistic model that incorporates signals from both linked-reads and paired-end reads into a unified model[15].

However, SV detection from linked-read datasets is still in the early stage. The available SV callers face challenges if we want to detect: (i) SVs from targeted region sequencing (e.g., WES); (ii) somatic SVs in cancer or somatic mosaic SVs that have low variant allele frequencies (VAFs, also known as variant allele

fractions); (iii) SVs of which the exact breakpoints have no coverage or are located in repeat regions. In this study, we introduce LinkedSV, a novel computational method and software tool for linked-read sequencing, which aims to address all the above challenges. LinkedSV detects large SVs using two types of evidence and quantifies the evidence using a novel probabilistic model. It also leverages read depth, paired-end signals and local assembly to detect small deletions. We evaluate the performance of LinkedSV on both whole-genome and WES data sets. In each case, LinkedSV outperforms other existing tools, including Longranger, GROC-SVs and NAIBR, especially on exome data and on somatic SVs with low variant allele frequencies. We additionally demonstrate clinical cases where LinkedSV identifies disease causal SVs from linked-read exome sequencing data missed by conventional exome sequencing, and show examples where LinkedSV identifies SVs missed by high-coverage long-read sequencing.

## Results

**Illustration of two types of evidence near SV breakpoints**. Two types of evidence may be introduced while a genomic rearrangement happens: (1) reads from one HMW DNA molecule which spans the breakpoint being mapped to two genomic locations and (2) reads from two distant genome locations that get mapped to adjacent positions. Both types of evidence can be used for SV detection.

First, we describe the signals of type 1 evidence. After reads mapping, the original HMW DNA molecules can be computationally reconstructed from the sequenced short reads using their barcodes and mapping positions. In order to distinguish them from the physical DNA molecules, we use fragments to refer to the computationally reconstructed DNA molecules. A fragment has a left-most mapping position, which we call L-endpoint, and a right-most mapping position, which we call R-endpoint. As a result of genomic rearrangement, reads from one breakpoint-spanning HMW DNA molecule would be mapped to two different genome loci on the reference genome. This split-molecule event has two consequences: (1) observing two separate fragments sharing the same barcode and (2) each of the two fragment has one endpoint close to the true breakpoints. Therefore, in a typical linked-read WGS data set, multiple split-molecule events could be captured and we would usually observe multiple shared barcodes between two distant genome loci and multiple fragment endpoints near the breakpoints.

To illustrate this, Fig. 1a shows the split-molecule events of a deletion, where breakpoints 1 and 2 are marked by red arrows. Multiple fragment endpoints are enriched near the two breakpoints of a large deletion. This can be observed in deletions with minimal size of about 5–10 kb. Figure 1b and Supplementary Figs. 1–3 show the patterns of enriched fragment endpoints that are introduced by different types of SVs. As an example, Fig. 1c shows the number of fragment endpoints in a 5-kb sliding window near two deletion breakpoints, based on a 35× coverage linked-read WGS data generated from the NA12878 genome (genome of a female individual extensively sequenced by multiple platforms). At the breakpoints, the number of fragment endpoints in the 5-kb sliding window is more than 100 and is 5 times more than normal regions, forming peaks in the figure.

Since the fragments can be paired according to their barcodes, we can also observe fragment endpoints of this deletion in a two-dimensional view. As shown in Fig. 1d, each dot indicates two endpoints from a pair of fragments which share the same barcode. The x-value of the dot is the position of the first fragment's R-endpoint and the y-value of the dot is the position of the second fragment's L-endpoint. The bottom panel and right

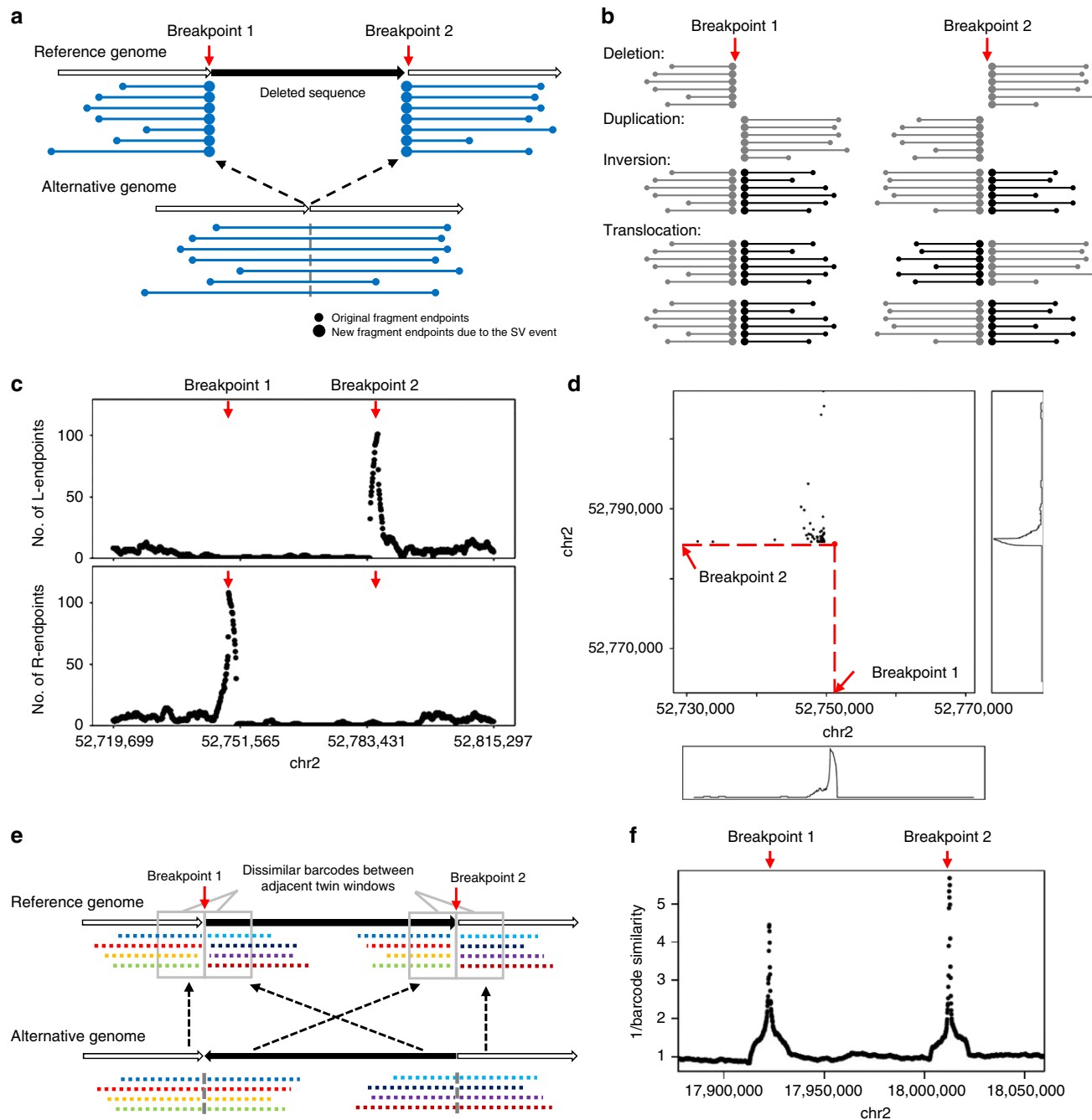

**Fig. 1 Two types of evidence near SV breakpoints. a** Type 1 evidence. Reads from HMW DNA molecules that span the breakpoints of a deletion are mapped to two genomic locations, resulting in two sets of observed fragments and two sets of newly introduced fragment endpoints (large dots). **b** The patterns of enriched fragment endpoints indicate the SV types. Please refer to Supplementary Figs. 1–3 and Supplementary Movie 1 for detailed explanations of how the patterns are formed. **c** Enriched fragment endpoints detected near two breakpoints of a deletion on NA12878 genome. L-endpoints and R-endpoints are plotted separately. The breakpoint positions are marked by red arrows. **d** Two-dimensional view of enriched endpoints near the two breakpoints of the deletion. Each dot indicates a pair of fragments which share the same barcode and thus may support the SV. The x-value of the dot is the position of the first fragment's R-endpoint and the y-value of the dot is the position of the second fragment's L-endpoint. The background of the 2D plot is cleaner than the 1D plot (panel c) since the fragments that do not share barcodes are excluded. **e** Type 2 evidence. Reads from two breakpoints of an inversion being mapped to nearby positions (in the gray rectangles), resulting in decreased barcode similarity between the two nearby positions. **f** Decreased barcode similarity near the breakpoints of an inversion on NA12878 genome. The reciprocal of barcode similarity is shown in the figure. The peaks indicate the positions of the breakpoint.

panel in Fig. 1d shows number of dots that are projected to the x-axis and y-axis. Similar with the one-dimensional plot (Fig. 1c), a peak is formed near each breakpoint, which is marked by the red arrow. The background noise of the two-dimensional plot is cleaner than the one-dimensional plot since the fragments that do

not share barcodes are excluded. Therefore, the two-dimensional plot is more useful when the variant allele frequency (VAF) is very low and there are only a few supporting fragments.

Next, we describe the signals of type 2 evidence. The barcodes between two nearby genome locations is highly similar because

the two locations are spanned by almost the same set of input HMW DNA molecules. However, due to the genome rearrangement, the reads mapped to the left side and right side of a breakpoint may originate from different locations of the alternative genome and thus have different barcodes (Fig. 1e). Dropped barcode similarity between two nearby loci therefore indicates an SV breakpoint. LinkedSV detects this type of evidence by a twin-window method, which uses two adjacent sliding windows to scan the genome and find regions where the barcode similarity between the two nearby window regions is significantly decreased. Figure 1f illustrates an inversion breakpoint detected by LinkedSV from the NA12878 genome. The change of barcode similarity was plotted and a peak was formed at the breakpoint. After searching for the two types of evidence, LinkedSV combines the candidate SV regions, and quantifies the evidence using a novel probabilistic model. The breakpoints are further refined using short-read information, including discordant read pairs and split-reads.

**Performance evaluation on simulated WGS data.** To assess LinkedSV's performance, we simulated a 35× linked-read WGS data set with 1175 SVs inserted using LRSIM[13] (see Methods for details). The breakpoints of the simulated SVs were designed to be located in repeat regions, since we found that LinkedSV and other available SV callers performed very well when the breakpoints were located in non-repeat regions, and thus we set to test the performances of all the SV callers under more challenging situations. This makes sense because SV breakpoints are more likely to be in repeat regions[7–9], and because these situations represent those that are difficult to be addressed by conventional short-read sequencing approaches.

The simulated reads were aligned to the reference genome using the Longranger[12] package provided by 10x Genomics. The Longranger pipeline internally uses the Lariat aligner[16], which was designed for the alignment of linked reads. SV calling was performed using LinkedSV as well as three other available SV callers designed for linked-read sequencing: Longranger, GROC-SVs[14], and NAIBR[15]. Two widely used short-read SV callers (Delly[3] and Lumpy[17]) were also used.

We used recalls, precisions and F1 scores to evaluate the performance of the six SV callers on this data set. As shown in Fig. 2a, the four linked-read SV callers showed higher F1 scores than the two short-read SV callers. LinkedSV achieved the highest recall and F1 score among all methods. GROC-SVs had a good precision but its recall was lower than LinkedSV, so we further analyzed the false negative calls of GROC-SVs to understand the underlying reason. A major portion of the false negative calls by GROC-SVs represents duplications that are smaller than twice the fragment length. For large duplications, the reads of the alternative allele are separated by a large gap so that we can observe two sets of fragments with the same set of barcodes, which indicate an SV (Supplementary Fig. 4a). If the duplication is not large enough, the reads will be probably clustered into one fragment (Supplementary Fig. 4b). Even in this case, we can observe enriched fragment endpoints near the duplication breakpoints in LinkedSV. As an example, Fig. 2b shows the endpoint signal of a missed duplication call by GROC-SVs. The supporting fragments of this duplication is shown in Fig. 2c. A detailed explanation of the pattern of duplication can be found in Supplementary Fig. 1 and Supplementary Movie 1. Figure 2d showed the extra read depth in this region. We also evaluated the breakpoint precision of LinkedSV. Most of breakpoints predicted by fragment endpoints are within 20 bp (Fig. 2e) and refined breakpoints using discordant read-pairs and split-reads have base-pair resolution (Fig. 2f).

**Benchmarking on WGS data with somatic SVs of low VAF.** Somatic SVs are commonly found in cancer genomes[18–20]. However, due to the high heterogeneity of genomic alteration in cancer genomes, somatic SVs often have low (as opposed to ~50% in a germline genome) VAF and thus are more difficult to detect by SV callers designed for germline SVs. We simulated two WGS data sets with VAF of 10% and 20%, respectively. Recalls, precisions and F1 values of the six SV callers were evaluated on both data sets (Fig. 3a, b). When the VAF was 20%, the recall of LinkedSV (0.803) was much higher than that of Longranger (0.306), GROC-SVs (0.324), and NAIBR (0.679) The F1 score of LinkedSV (0.855) was also the highest among all the SV callers. When the VAF was 10%, LinkedSV still had a recall of 0.761, which was 72% higher than the second best SV caller NAIBR. Longranger detected 17% of the SVs while GROC-SVs almost completely failed to detect the SVs. The recall rates of Delly and Lumpy were 0.28 and 0.72, respectively, indicating that some of the SVs can be detected even without barcode information. These observations confirmed that other SV callers were mainly designed for germline genomes and had substantial difficulty in detecting SVs with somatic mosaicism. However, due to the combination of barcode overlapping and enriched fragment endpoints in our statistical model (see Methods for details), LinkedSV was able to achieve a good performance even when VAF was very low. We manually checked the barcode overlapping evidence of some SV calls using the Loupe software developed by 10x Genomics. Figure 3c shows an inversion that was missed by Longranger, and NAIBR but detected by LinkedSV (at VAF of 10%). Although the variant frequency is low, the overlapped barcodes between the two inversion breakpoints can be clearly visualized (in the black circles) in the figure. Figure 3d shows the supporting fragments of the inversion detected by LinkedSV. Each horizontal line represent two fragments that share the same barcode and support the SV. These results suggest that the manufacturer-provided software tool has limitations for SV detection, despite its strong functionality in visualization.

To test the performance of LinkedSV on the detection of disease casual SVs, we simulated one germline and two somatic (VAF = 10% and 20%) linked-read WGS data sets with 51 deletions/duplications that were known to cause copy number variation (CNV) syndromes involved in developmental disorders (see Method for details). The size distribution of the events was shown in Supplementary Fig. 5. The performances of LinkedSV as well as five other SV callers were shown in Supplementary Fig. 6. The results were similar to those of the above simulations. LinkedSV had the highest F1 score on both germline and mosaic data sets, followed by NAIBR.

**Benchmarking of deletion detection on the HG002 genome.** Recently, the Genome in a Bottle (GIAB) Consortium released a benchmark call set for the evaluation of germline SV detection[21]. The benchmark set was based on the HG002 genome and was generated from integrating multiple SV calling methods from multiple sequencing platforms including 10x Genomics sequencing and PacBio long-read sequencing. The current GIAB call set only contains insertions and deletions. Since LinkedSV and the other three linked-read SV callers cannot detect insertions, we only benchmarked the performance to detect deletions using this benchmarking data set.

LinkedSV uses different strategies to detect deletions of different sizes. For deletions that are more than 10 kb, LinkedSV uses the two types of evidence from barcode signals as described above; for deletions that are within 1–10 kb, LinkedSV uses a combination of read depth and paired-end signals, with additional consideration of local haplotypes; for detection of

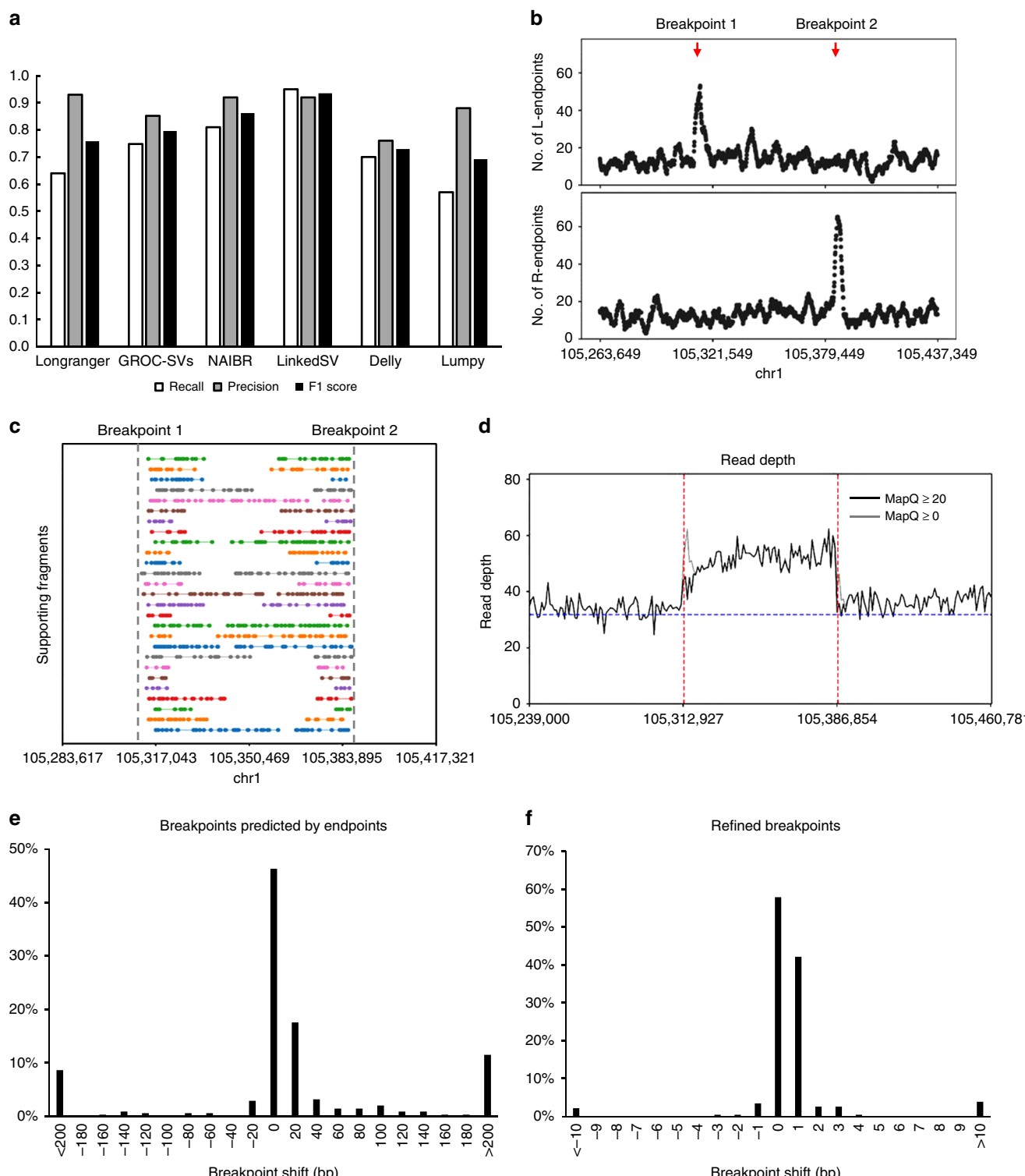

**Fig. 2 Performance of LinkedSV on the simulated WGS data set. a** Recalls, precisions and F1 scores of six SV callers on the simulated WGS data set. **b** Fragment endpoint signals of a small duplication that was missed by GROC-SVs. The peaks indicate the approximate breakpoint positions. **c** Supporting fragments of the tandem duplication. These are fragments span the junction of the first copy and the second copy. Please refer to Supplementary Fig. 1 and Supplementary Movie 1 for detailed explanations of how the patterns are formed. Horizontal lines represent linked reads with the same barcode; dots represent reads; colors indicate barcodes; dashed vertical gray lines represent breakpoint positions. **d** Read depth distribution near the duplication region. The black lines showed the depth of reads with mapping quality ≥ 20, while the gray lines showed the depth of reads with mapping quality ≥ 0 (i.e., all reads). Red lines indicate breakpoints predicted by LinkedSV and the blue line indicate the average depth of the whole genome. **e** Precision of breakpoints predicted by LinkedSV without checking short-read information. **f** Precision of LinkedSV refined breakpoints using discordant read-pairs and split-reads. Source data is provided as a Source Data file.

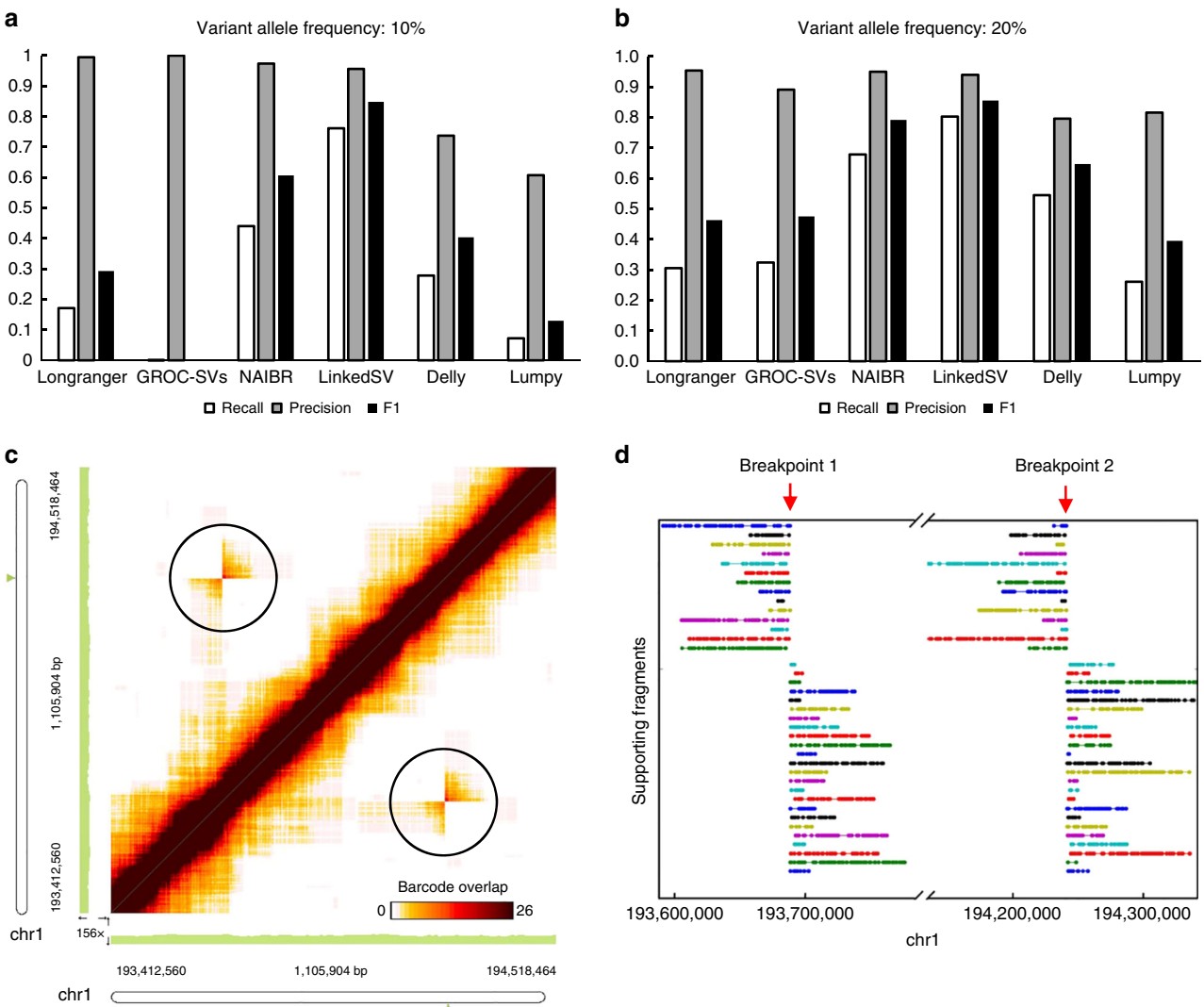

**Fig. 3 Performance of LinkedSV on the simulated WGS data with low variant allele frequencies. a**, **b** Recalls, precisions and F1 scores of six SV callers on the simulated WGS data set with VAF of 10% and 20%. **c** Heap map of overlapping barcodes in chr1:193412560–194518464 (hg19 coordinates) showing an inversion that was missed by Longranger, and NAIBR (VAF = 10%). The overlapping barcodes between the two inversion breakpoints can be clearly visualized (in the black circles). The heat map was plotted by the Loupe software (10x Genomics). Dots represent overlapping barcodes. **d** Supporting fragments of the inversion detected by LinkedSV. Horizontal lines represent linked reads with the same barcode; dots represent reads; colors indicate barcodes. Predicted breakpoint positions are marked by red arrows. Source data is provided as a Source Data file.

SVs that are less than 1 kb, LinkedSV uses a local assembly based method. Specifically, we modified the FermiKit[22] de novo assembly pipeline to be a local assembler to improve speed and reduce the complexity of the assembly graph (see Method for details).

Supplementary Fig. 7a showed the performance on detection of deletions that were more than 10 kb. The recall and F1 score of LinkedSV was the highest among the seven methods. The four linked-read SV callers performed better than the three short-read SV callers in terms of F1 score. The performance on detection of deletions that within 1–10 kb were shown in Supplementary Fig. 7b. The performance of LinkedSV was similar to Longranger, which also provided an algorithm to detect small deletions. NAIBR and GROC-SVs did not perform well because they were not designed to detect small events including small deletions. For deletions that were less than 1 kb, LinkedSV (with modified FermiKit) performed best (recall = 0.48, F1 score = 0.64), it detected more calls than the original de novo assembly version (recall = 0.43, F1 score = 0.60), indicating that local assembly reduced the complexity of the assembly graph and improved the

performance. NAIBR, GROC-SVs, and Lumpy did not perform well on deletions of this scale (Supplementary Fig. 7c). Size distribution of SV events (including deletions, duplications, and inversions) detected by LinkedSV was shown in Supplementary Fig. 8.

**Performance evaluation on simulated WES data.** Compared with WGS, WES is currently widely used in clinical settings to identify disease causal variants on patients with suspected genetic diseases, partly due to the lower cost of WES. Since WES only covers a small portion of regions in the whole genome, it is far more challenging to detect SVs from WES data, especially when the SV breakpoints are not in the capture regions. However, by combining linked-read sequencing with WES capture platforms, it is possible to alleviate this problem, and significantly improve the sensitivity of SV detection using WES.

To evaluate SV detection on linked-read WES data, we simulated a 40× coverage linked-read WES data set with 1160 heterozygous SVs (see Methods for details). Totally, 44.3% of the

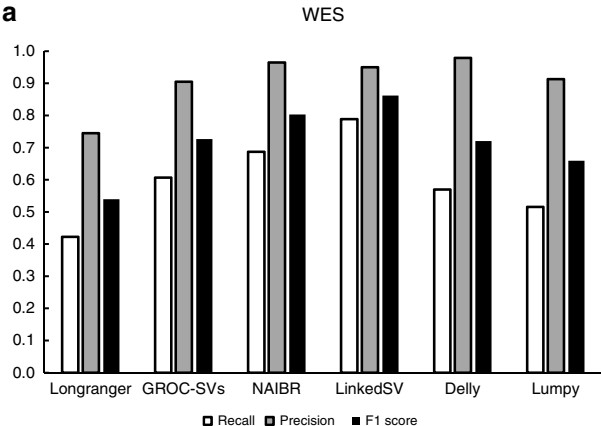

**a**

WES

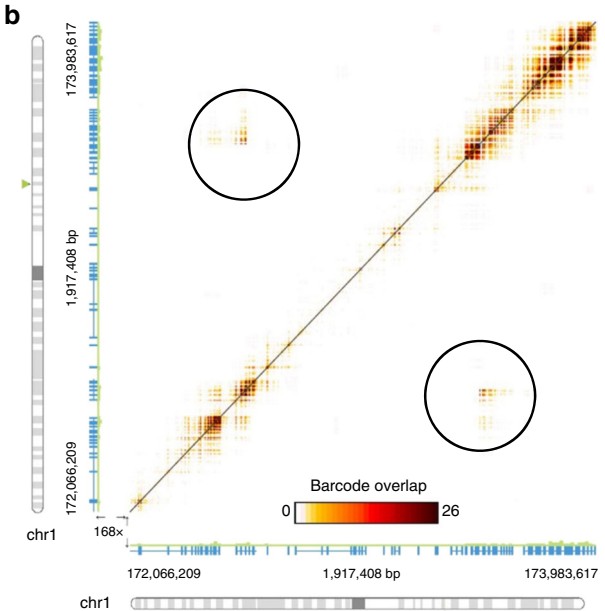

**b**

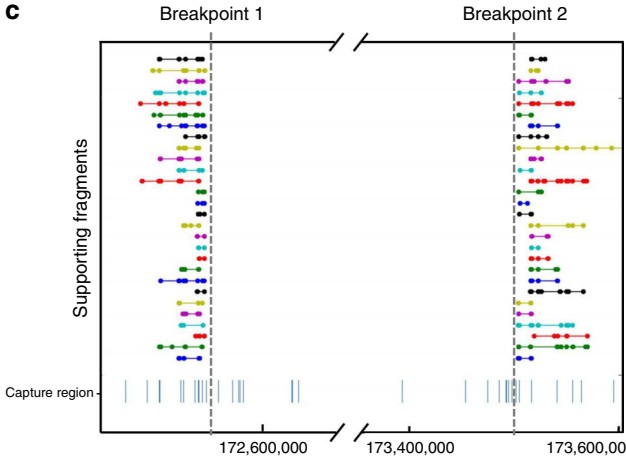

**c**

**Fig. 4 Performance of LinkedSV on the simulated WES data set. a** Recalls, precisions and F1 scores of six linked-read SV callers on the simulated WES data set. **b** Heat map showing a deletion that was missed by NAIBR. The overlapping barcodes between the two breakpoints can be clearly visualized (in the black circles). The heat map was plotted by the Loupe software. Dots represent overlapping barcodes. **c** Supporting fragments of the deletion detected by LinkedSV. Horizontal lines represent linked reads with the same barcode; dots represent reads; colors indicate barcodes. Predicted breakpoint positions are marked by vertical gray lines. Capture regions were shown as vertical bars in the bottom. Source data is provided as a Source Data file.

barcodes but lack short-read support. For example, Fig. 4b showed a deletion between chr1:172545561–173504265. Both breakpoints were located outside of capture regions. Breakpoint 1 (chr1:172545561) was 768 bp away from the nearest capture region and breakpoint 2 (chr1:173504265) was 392 bp away from the nearest capture region. Unfortunately, no discordant read pairs that support the deletion could be found. However, shared barcodes between the two breakpoints were clearly indicated by the Loupe software (Fig. 4b). In addition, LinkedSV also detected 28 pairs of fragments that share the same barcodes and support the SV. These fragments were plotted in Fig. 4c. Although no short-read support was found, the SV type could be determined using the pattern of enriched fragment endpoints shown in Fig. 1b. In this SV event, R-endpoints were highly enriched for the first set of fragments and L-endpoints were highly enriched for second set of fragments. Thus, the SV type was predicted as deletion.

**Detection of *F8* inversion from clinical WES data**. We also tested the performance of LinkedSV on several clinical samples with linked-read WES data. First, we applied LinkedSV on a WES sample of a male individual with Hemophilia A. Previous experiments had shown that the patient had type I inversion of the *F8* gene, where the two breakpoints resided in intronic/intergenic regions, thus the inversion and its breakpoints cannot be inferred from conventional WES. The *F8* gene is located in Xq28. The intron 22 of *F8* gene contains a GC-rich sequence (named int22h-1) that is duplicated at two positions towards the Xq-telomere (int22h-2 and int22h-3). Int22h-2 has the same direction with int22h-1 while int22h-3 has the inverted direction. The type I inversion is induced by the recombination between int22h-1 and int22h-3[23,24] (Fig. 5a). BLAST alignment of int22h-1 and int22h-3 showed that the two sequences had 99.88% identity. Since the breakpoints were located in two segmental duplications with nearly identical sequences, the inversion is undetectable by conventional short-read sequencing. Delly[3] and Lumpy[17] failed to detect the inversion from the linked-read WES data (results were shown in Supplementary Tables 1 and 2).

Longranger, GROC-SVs, NAIBR, and LinkedSV were also used to detect SVs from this sample. None of the first three methods detected this inversion (results were shown in Supplementary Tables 3–5), although the overlapping barcodes can be visualized using the Loupe software (Fig. 5b). However, LinkedSV successfully detected this inversion by combining two types of evidence. As described above, barcode similarity between two nearby regions is very high but drops suddenly at the breakpoints. Figure 5c shows the sudden drop of barcode similarity at the two breakpoints. Each dot in the figure represents the reciprocal of the barcode similarity between its left 40 kb window and right 40 kb window, thus the y-value of the dots are inversely related to the barcode similarity and positively related to the probability of being a breakpoint. The barcode similarities are lowest at the two breakpoints and thus form two peaks in the figure (marked by red arrow). In addition, LinkedSV also identified the supporting

breakpoints were not in exon regions. SV calling was performed using the six SV callers. As shown in Fig. 4a, LinkedSV had the highest recall (0.79) and highest F1 score (0.86). In terms of the balanced accuracy (F1 score), NAIBR was the second best caller, followed by GROC-SVs.

We analyzed false-negative calls of the second best SV caller NAIBR. NAIBR tends to miss some SV events that have shared

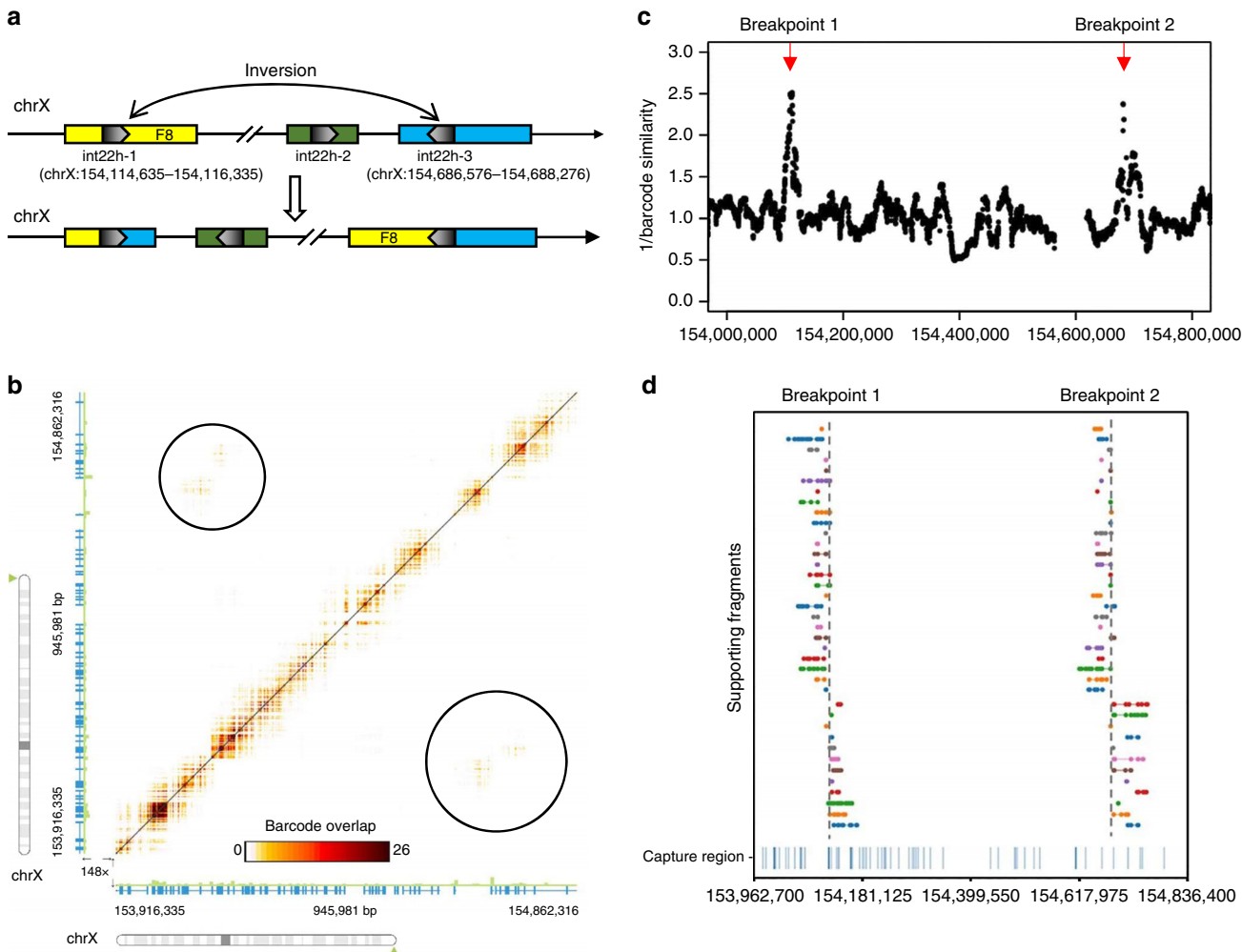

**Fig. 5 Detection of *F8* inversion from clinical exome sequencing data. a** Illustration of type I inversion of *F8* gene. A portion of intron 22 has three copies in chrX (int22h-1, int22h-2, and int22h-3). The inversion is induced by the homologous recombination between two inverted copies int22h-1 and int22h-3. Int22h-1 is located in intron 22 of *F8* gene and int22h-3 is located in the intergenic regions. **b** Heat map of overlapping barcodes in chrX:153916335-154862316 (hg19 coordinates), plotted by the Loupe software tool. Black circles indicate overlapping barcodes near the inversion breakpoints. Dots represent overlapping barcodes. **c** Decreased barcode similarity at breakpoints detected by the twin window method of LinkedSV. Window size = 40 kb. **d** Supporting fragments detected by LinkedSV. Horizontal lines represent linked reads with the same barcode; dots represent reads; colors indicate barcodes. Dashed vertical gray lines represent breakpoints. Capture regions were shown as vertical bars in the bottom.

fragments of the SV using type 1 evidence (Fig. 5d). The predicted breakpoint positions are consistent with the genomic positions of int22h-1 and int22h-3.

**Detection of mosaic *NF1* deletion from clinical WES data.** Another linked-read WES sample was from an individual who was clinically diagnosed with Neurofibromatosis type 1. Neurofibromatosis type 1 is caused by mutations in the *NF1* gene on chromosome 17q11.2, which encodes neurofibromin, a GTPase activating protein that has a role in the regulation of RAS signaling. Since standard genetic testing techniques including cDNA sequencing and multiplex ligation-dependent probe amplification revealed no constitutional or mosaic pathogenic mutation in this patient, we hypothesized that this patient may carry an SV affecting the *NF1* gene that escapes the detection by the applied standard techniques. To evaluate LinkedSV, we utilized the 10x Genomics Chromium platform to generate linked-read WES data to confirm and resolve the mutation. SV detection was conducted using the four linked-read SV callers as well as Delly and Lumpy. Longranger detected overlapped barcodes between exon 54 of the *NF1* gene and intron 3 of *RAB11FIP4*. However, the SV type was

unknown and no supporting read pairs or split-reads were found. GROC-SVs, NAIBR, Delly and Lumpy failed to detect this SV (Supplementary Tables 6–9). As shown in Fig. 6a, LinkedSV detected 16 fragment pairs that may support a deletion spanning the region of chr17:29684175–29822527. In addition, a discordant read pair spanning the two breakpoints were found (Fig. 6b), which gave further evidence supporting the deletion. The breakpoints were estimated from this discordant read pair and thus the resolution is a few hundred base pairs. In Fig. 6, each colored line represent a reconstructed fragment, and ~13% of the fragments belong to the variant allele, indicating the somatic mosaicism of this deletion. The right breakpoint was within an AluJr sequence masked by repeat masker, which may explain why the deletion was difficult to be detected by conventional methods.

In comparison, the clinical lab used massive parallel sequencing (TruSightCancer panel on a MiSeq platform (Illumina)) and successfully revealed in exon 54 a transition of *NF1* sequences into a non-*NF1* derived sequence. This sequence transition at *NF1* position c.7886_7887 was present in 8% of the reads covering this site in germline DNA of the patient. Analysis of the reads displaying the aberrant sequence in exon 54 showed that the

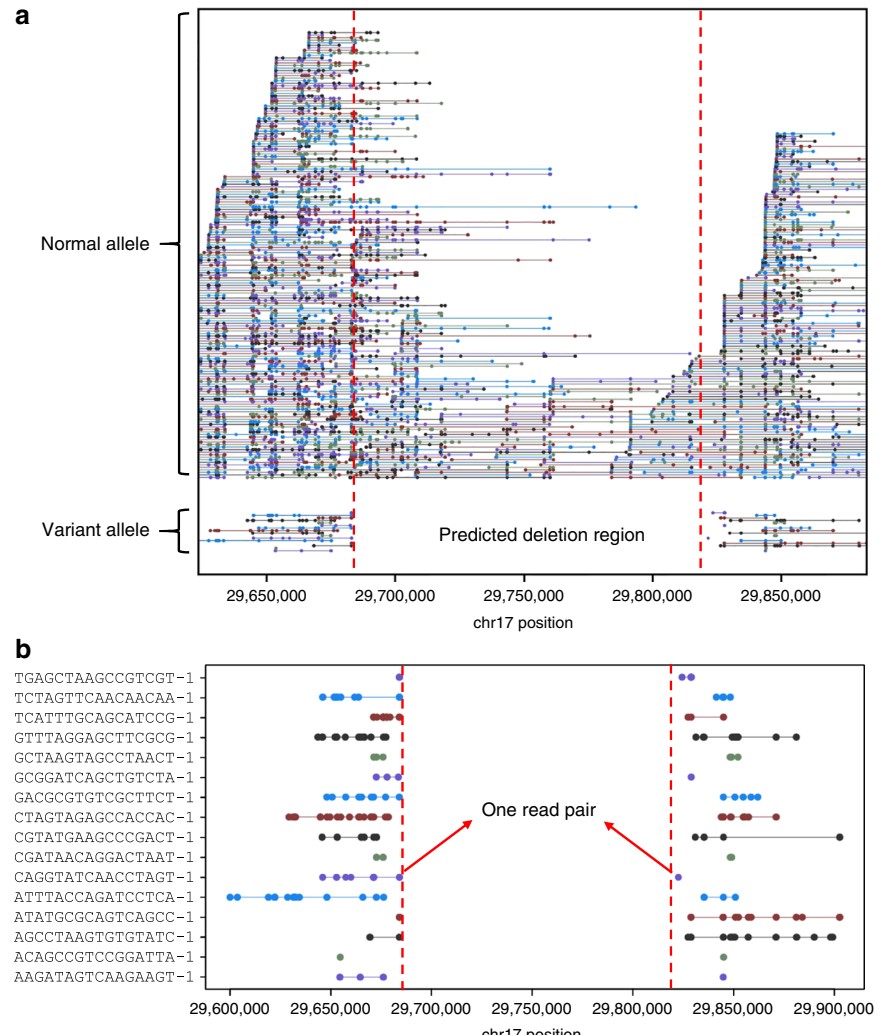

**Fig. 6 Detection of *NF1* deletion from clinical exome sequencing data. a** Plot of linked-reads for *NF1* WES sample spanning chr17:29645000–29855000. In the normal allele (top), there are 71 fragments crossing over the left breakpoint and 38 fragments crossing over the right breakpoint. In the variant allele (bottom), the linked reads are separated by a large gap. Horizontal lines represent linked reads with the same barcode; dots represent reads; colors indicate barcodes. Dashed vertical red lines represent breakpoints. **b** Zoom-in plot of supporting fragments for the deletion. One read pair was found to support the deletion.

non-*NF1* derived sequence was part of an Alu element that matched best a sequence in intron 3 of the *RAB11FIP4* gene located 138 kb downstream of *NF1* exon 54. These results suggested a low-level (~8%) mosaicism of a deletion encompassing the region intervening between *NF1*:c.7886 and *RAB11FIP4*:c.337–22216, so that the true deletion spans chr17:29684367–29822453, which is very similar to our estimated breakpoints from LinkedSV above. In summary, our analysis on two clinical samples with *F8* inversion and *NF1* deletion demonstrated the unique advantage of linked-read sequencing in confirming and resolving SVs in repetitive regions and challenging situations.

**Comparison with SVs detected from long-read sequencing**. We previously reported the de novo genome assembly of a Chinese individual (HX1)[25]. This genome was sequenced deeply at 103× coverage using PacBio long-read sequencing. Recently, the developers of SMRT-SV[10,26] reported the SV calls of HX1 detected from the PacBio data. In addition, we have also generated a 37× linked-read WGS dataset on HX1. Therefore, in the current study, we detected SVs from the linked-read data using LinkedSV and compared the SV calls detected by LinkedSV and SMRT-SV. The SMRT-SV call set has 17 large deletions (≥10 kb), all of which were detected by LinkedSV. In addition, LinkedSV

detected another 46 large deletions, which were missed by SMRT-SV. To validate these deletion calls, we mapped the PacBio reads of HX1 to GRCh38 reference genome using minimap2[27], and manually examined all the SV-affected regions in both PacBio data and linked-read data, using the Integrative Genomics Viewer (IGV)[28] and the Loupe software tool. We classify a deletion as a true deletion if there are decreased read depth in the deletion region and clear boundaries at the breakpoints. After the manual inspection, we found that among the 46 deletions that are only detected by LinkedSV, 34 of them have clear evidence of deletion in the WGS data; 10 of them are complex SV events that need to be fully resolved; and 2 of them are false positive events. Figure 7a–c showed an example of a deletion that were detected by LinkedSV but missed by SMRT-SV. This is a 45 kb deletion located in chr2:110395971–110441346. A deletion pattern was clearly indicated by the Loupe software tool (Fig. 7a). After examine the PacBio reads, we were able to found clipped reads at the breakpoint positions (Fig. 7b, c). However, for most of the clipped reads, the clipped sequences were aligned to the hs38d1 decoy sequence, except for 5 reads with clipped sequence > 7 kb. Analysis of the 5 reads revealed that the two breakpoints in chr2 were not directly joined. There was a 6 kb insertion in between. The inserted sequence was from hs38d1

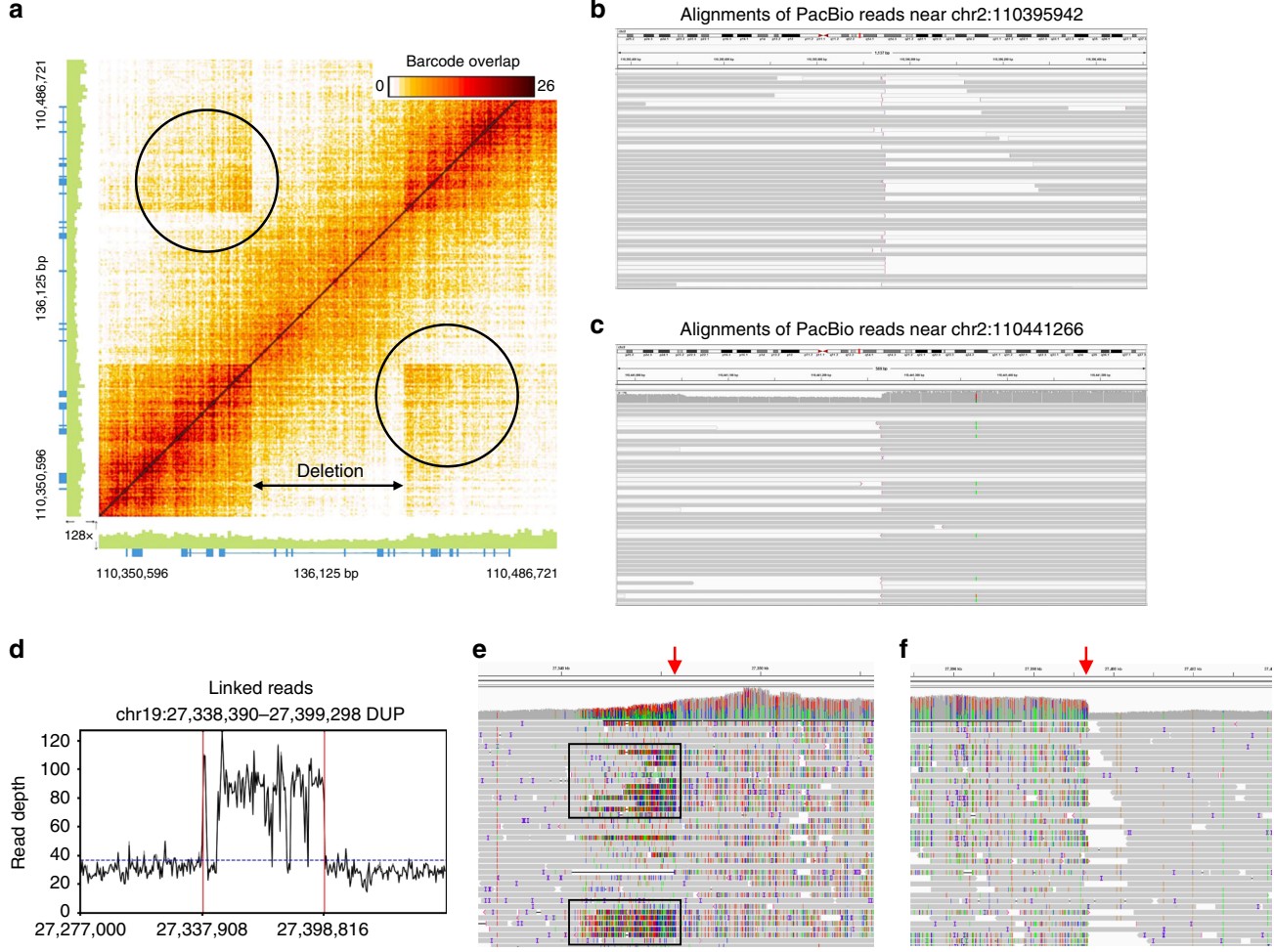

**Fig. 7 Structural variants detected from linked-read WGS data of the HX1 genome. a** Heatmap of overlapping barcodes for a 45 kb deletion on chromosome 2 (chr2:110395971–110441346, hg38 coordinates), plotted by the Loupe software tool. Black circles indicate overlapping barcodes near the breakpoints. The deletion was not detected by SMRT-SV from PacBio long reads; **b**, **c** Alignments of PacBio reads near the breakpoints of the 45 kb deletion in chr2 in the HX1 genome. Clipped reads were marked by vertical pink lines (5'-clipping) or pink arrows (3'-clipping). The figures were generated by IGV. Reads with mapping qualities equal to 0 were in white color. **d** Read depth distribution near a 61 kb duplication region on chromosome 19 (chr19:27338390–27399298, hg38 coordinates). The calculation was based on the bam file of linked-reads. Only reads with mapping quality ≥20 were counted. The dotted blue line showed the average depth across the whole genome. The predicted breakpoints were indicated by vertical red lines. The duplication was not detected by SMRT-SV using PacBio long reads. **e**, **f** Aligned PacBio raw reads near the two breakpoints of the duplication, as shown in IGV. Increased alignment mismatches due to the SV were observed in (**e**) (black rectangles). A clear duplication breakpoint was observed in (**f**).

(coordinates: 1381394–1387327). The proposed variant allele was shown in Supplementary Fig. 9a. To validate this deletion/insertion event, we aligned all the PacBio reads to a new reference genome with all sequences of GRCh38 plus hs38d1 and the sequence of the proposed variant allele. The reads aligned to the proposed variant allele were shown in Supplementary Fig. 9b. There were 33 reads spanning the chr2-hs38d1 junction, 48 reads spanning the hs38d1-chr2 junction and 13 reads spanning both junctions. De novo assembly of all the reads aligned to the proposed variant allele generated a single contig of 42.7 kb, which also spanned both junctions (Supplementary Fig. 9b, bottom track). These analysis showed that the large deletion event detected by LinkedSV is true and with PacBio long reads the details of complex SV events could be resolved.

We also compared the duplication calls of LinkedSV and SMRT-SV and manually examined discordant SV calls. LinkedSV reported 6 large duplications (≥10 kb), 5 of which were not reported by SMRT-SV. Figure 7d–f showed the evidence of a 61 kb duplication call (chr19:27338390–27399298), which was only reported by LinkedSV. A two-fold increase of read depth could be

observed in the duplication region (Fig. 7d), and the breakpoints were also clearly indicated in the alignments of PacBio long reads, as shown in IGV (Fig. 7e, f). The read depths of PacBio raw reads and error-corrected reads were shown in Supplementary Fig. 10. The increase of read depth in the duplication region can also be observed. After the manual inspection of the left duplication breakpoint, a small duplication event was found next to the main event. The boundaries of the small duplication can be observed in the alignments of linked reads and error-corrected PacBio reads, but not in the alignments of PacBio raw reads, potentially because of mapping errors (Supplementary Fig. 11). SMRT-SV reported 194 large duplications (≥10 kb). Unexpectedly, 193 of the duplication calls were not detected by LinkedSV. In addition, none of these duplications could be detected by Sniffles[11], another widely used long-read SV caller. After comparing with the segmental duplication database[29], we found that 182 of the 193 duplication calls (94.3%) were located in large segmental duplication regions. Both long reads and linked reads could not be reliably mapped in these regions. As an example, we plotted the read depth distribution in the region around a 25 kb

duplication call of SMRT-SV. Neither long reads (Supplementary Fig. 12a) nor linked reads (Supplementary Fig. 12b) had mapping quality >20 in the duplication region. Therefore, SV detection in the super-large segmental duplication regions is still very challenging. In summary, our comparative analysis demonstrated unique advantages of linked-read WGS in resolving large SVs that may fail to be detected by even long-read sequencing platform with very deep coverage.

## Discussion

In this study, we present LinkedSV, a novel algorithm for SV detection from linked-read sequencing data. We assessed the performance of LinkedSV on three simulated data sets and two real data sets. By incorporating the two types of evidence as outlined below, LinkedSV outperforms all existing linked-read SV callers including Longranger, GROC-SVs, and NAIBR on both WGS and WES data sets.

Type 1 evidence gives information about which two genomic positions are connected in the alternative genome. It has two observations: (1) fragments with shared barcodes between two genomic locations and (2) enriched fragment endpoints near breakpoints. Current existing linked-read SV callers only use the first observation to detect SVs, while LinkedSV incorporates both observations in the statistical model and is therefore more sensitive and can detect SVs with lower allele frequencies, such as somatic SVs in cancer genomes and mosaic structural variations.

Type 2 evidence gives information about which genomic position is interrupted with the observation that the reads on the left side and right side of a genomic position have different barcodes and should be derived from different HMW DNA molecules. LinkedSV is the only SV caller that use type 2 evidence to detect breakpoints. Type 2 evidence is independent of type 1 evidence, and gives additional confidence to identify the breakpoints. In addition, type 2 evidence can be detected locally, which means we can detect an abnormal genomic location without looking at the barcodes of the other genomic locations. This is particularly useful in two situations: (1) novel sequence insertions where there is only one breakpoint and (2) only one breakpoint is detectable and the other breakpoint is located in a region where there is little coverage within 50 kb, which is often the case in target region sequencing. As LinkedSV incorporates two types of evidence from barcodes, and performs local assembly to detect small deletions, the computation time of LinkedSV is longer than NAIBR, but shorter than GROC-SVs and Longranger (Supplementary Fig. 13).

In recent years, WES has been widely used to identify disease causal variants for patients with suspected genetic diseases in clinical settings. Identification of SVs from WES data sets are more challenging because the SV breakpoints may not be in the capture regions and thus there would be little or no coverage at the breakpoints. Linked-read sequencing increases the chance of resolving such type of SVs by providing long-range information. As long as there are a few capture regions nearby, the fragments can still be reconstructed and type 1 and type 2 evidence can still be observed. Our statistical models for both type 1 evidence and type 2 evidence were designed to handle both WGS and WES data sets. GROC-SVs uses a local-assembly method to verify the SV call, which requires sufficient coverage at the breakpoints. By using these two types of evidence, LinkedSV can rely less on short-read information (e.g., pair-end reads and split-reads). We demonstrated that LinkedSV has better recall and balanced accuracy (F1 score) on the simulated WES data set and can detect SVs even when the breakpoints were not located in capture regions and have no short-read support. In addition, LinkedSV is also the only SV caller that

clearly detected the *F8* intron 22 inversion and *NF1* deletion from the clinical WES data sets.

Linked-read sequencing has several advantages over traditional short-read sequencing for the purpose of SV detection. First, the human genome is highly repetitive. Previous studies have shown that SVs are closely related to repeats and many SVs are directly mediated by homologous recombination between repeats[30]. In traditional short-read sequencing, if the breakpoint falls in a repeat region, the supporting reads would be multi-mapped and thus the SV cannot be confidently identified. However, this type of SVs are detectable by linked-read sequencing when the HMW DNA molecules span the repeat region. We can observe type 1 and type 2 evidence in the non-repeat region nearby. In our benchmarking, LinkedSV detected more SVs than Delly and Lumpy, especially when the VAF is low. Secondly, SVs are undetectable from traditional short-read sequencing if there is little coverage at the breakpoints, which is often the case in WES data sets. As described above, this type of SV can also be resolved by linked-read sequencing and LinkedSV. Third, linked-read sequencing requires less coverage for detection of SVs with low variant allele frequencies. In linked-read sequencing data, short-read pairs are sparsely and randomly distributed along the HMW DNA molecule. In a typical linked-read WGS data set, the average distance between two read pairs derived from the same HMW DNA molecule is about 1000 bp and each HMW DNA molecule only has a short-read coverage of about 0.2×. Therefore, there are about 150 HMW DNA molecules (reconstructed fragments) covering a genomic location of 30× depth. An SV of 10% VAF will has 15 supporting fragment pairs in a 30× depth location in linked-read WGS data set, which is sufficient to be detected by LinkedSV. However, an SV of 10% VAF will only has 3 supporting read pairs in a 30× depth location in traditional short-read WGS, which makes the detection more challenging.

Linked-read sequencing also has several advantages over long-read sequencing in terms of SV detection. The fragment length of linked-reads (typically 50–100 kb) is longer than the average read length of regular long-read sequencing (typically 20–30 kb). Therefore, linked-read sequencing has unique advantages for detection of large SVs. In our study, LinkedSV detected several large SVs that were missed in the long-read SV call set. We also showed that the sequencing error (13–15%) of long-read sequencing technologies potentially had a negative effect on reads mapping and subsequent SV calling (Supplementary Fig. 11). In terms of library preparation, linked-read sequencing only requires 1 ng input DNA, which is two orders of magnitude smaller than what is needed by long-read sequencing. Therefore, disease samples of very low DNA amount can be easily sequenced by linked-read sequencing. In addition, SNPs, indels and SVs can be detected from linked-read sequencing simultaneously.

LinkedSV may have limitations on detection of SVs in large segmental duplication regions, where the linked reads have low mapping qualities. SMRT-SV was able to find 194 large duplications in the HX1 genome, which were not detected by LinkedSV and Sniffles, two alignment-based SV callers. SMRT-SV detects SVs using an assembly based approach. During the assembly process, the assembly contigs were error corrected and polished by the PacBio reads. Therefore, the assembly contigs are potentially more accurate and longer than each of the raw reads. Thus, it is possible for SMRT-SV to detect SVs in these large segmental duplication regions.

The linked-read technology provides strong evidence to detect large SVs, but it provides little additional evidence to detect small SVs. Therefore, LinkedSV has limited power to detect small SVs such as small duplications and inversions. However, based on our analysis of SV size distribution, large SVs are associated with diseases such as cancers and CNV syndromes (Supplementary

Note 2). Therefore, we expect that linked-read technology can help resolve disease associated SVs. Similar to the existing linked-read SV callers, LinkedSV currently does not handle insertions and repeat expansions. As a future direction, we plan to detect novel sequence insertions using type 2 evidence, since this type of SV also cause a decrease of barcode similarity between nearby regions and can be detected by the twin-window method. The exact insertion sequence may then be inferred from the assembly of all the reads that share barcodes with the candidate breakpoint. LinkedSV currently already supports local assembly to detect deletions, but it has not been parameterized and optimized to be combined with type 2 evidence for detection of insertions.

In summary, we present LinkedSV, a novel SV caller for linked-read sequencing. LinkedSV outperformed current existing SV callers, especially for identifying SVs with low allele frequency or identifying SVs from target region sequencing such as linked-read WES. We expect that LinkedSV will facilitate the detection of SVs from linked-read sequencing data and help solve negative clinical cases from conventional short-read sequencing.

## Methods

**Breakpoint detection from type 1 evidence**. First, LinkedSV reconstructs the original long DNA fragments from the reads using mapping positions and barcode information. All mapped reads are partitioned according to the barcode and sorted by mapping position. We define gap distance as the distance between two nearest reads with the same barcode. Two nearby reads are considered from the same long DNA fragment if they have the same barcode and their gap distance is less than a certain distance $G$. $G$ is determined using two steps. First, we use $G = 50$ kb (the same as Zheng et al.[12]) to group the reads into fragments. This value is suitable for detection of large SVs. However, it may be too large for detection of SVs that are smaller than 50 kb. Therefore, we calculate the empirical distribution of intra-fragment gap distance, which is the distance of two nearby reads that are grouped in one fragment. The empirical distribution of intra-fragment gap distance is calculated from all the fragments, and we assign $G$ as the 99th percentile of this distribution. $G$ is a fixed number for all fragments and is usually between 5 and 15 kb, depending on the data set. Fragments with a gap distance larger than $G$ potentially span a breakpoint and will be separated into two fragments.

In non-SV regions, all the reads from the same HMW DNA molecule would be reconstructed into a single-DNA fragment. The reads from the breakpoint-spanning HMW DNA molecule will be mapped to two different positions in the genome. As illustrated in the Result section, this split-molecule event has two consequences: (1) observing two fragments sharing the same barcode and (2) each of the two fragment has one endpoint close to the breakpoints. Therefore, we could observe enriched fragment endpoints near the breakpoints, in both one-dimensional view (Fig. 1c) and two-dimensional view (Fig. 1d). The type of the endpoints (L-endpoint or R-endpoint) that enriched near the breakpoints depends on the type of SVs (Fig. 1b). The two-dimensional view has less background noise because the fragments that do not share barcodes and thus do not support the SVs are excluded. Therefore, we detect the enriched endpoints in the two-dimensional view.

We now describe how we detect the type 1 evidence of deletion calls, but the method can be applied to other types of SVs. We define fragment pair to be two fragments sharing the same barcode. Let $b_1$, $b_2$ be the positions of the two breakpoint candidates (assuming $b_1 < b_2$). Let $n$ be the number of fragment pairs that may support the SV between $b_1$ and $b_2$. Let $F_{i1}$, $F_{i2}$ denote the $i$th fragment pair that support the SV. Let $B(F)$ denote the barcode of fragment $F$. Therefore, we have

$$B(F_{i1}) = B(F_{i2}), \ i = 1, 2, 3, \ldots, n. \tag{1}$$

Let $L(F)$ denote the L-endpoint position (i.e., left-most position) of fragment $F$, $R(F)$ denote the R-endpoint position (i.e., right-most position) of fragment $F$. Since this is a deletion and $b_1 < b_2$, $R(F_{i1})$ is the position on $F_{i1}$ that is closest to $b_1$ and $L(F_{i2})$ is the position on $F_{i2}$ that is closest to $b_2$ (Supplementary Fig. 14a). The distance between the fragment endpoint and its corresponding breakpoint should be within gap distance distribution (explained in Supplementary Fig. 15). Therefore, for almost all (99% × 99%) of the fragment pairs, we have

$$b_1 - G \le R(F_{i1}) \le b_1; \ b_2 \le L(F_{i2}) \le b_2 + G. \tag{2}$$

As described above, $G$ is the 99th percentile of the empirical distribution of intra-fragment gap distance.

If we regard $(R(F_{i1}), L(F_{i2}))$ as a point in a two-dimensional plane, according to Eq. (2), for almost all (98.01%) of the fragment pairs $(F_{i1}, F_{i2})$, $((R(F_{i1}), L(F_{i2}))$ is restricted in a $G \times G$ square region with the point $(b_1, b_2)$ being a vertex (Supplementary Fig. 14b).

We used a graph-based method to fast group the points into clusters and find square regions where the numbers of points were more than expected. First, every possible pair of endpoints $(R(F_1), L(F_2))$ meeting $B(F_1) = B(F_2)$ formed a point in the two-dimensional plane. Each point indicated a pair of fragments that share the same barcode. For example, if ten fragments share the same barcode, $C_{10}^2$ pairs of endpoints will be generated. A point/pair of endpoints may or may not support an SV because there are two possible reasons for observing two fragments sharing the same barcode: (1) the two fragments originated from two different HMW DNA molecules but were dispersed into the same droplet partition and received the same barcode and (2) the two fragments originated from the same HMW DNA molecule but the reads were reconstructed into two fragments due to an SV. The points are sparsely distributed in the two-dimensional plane and it is highly unlikely to observe multiple points in a specific region. Next, a $k$–$d$ tree ($k = 2$) was constructed, of which each node stores the $(X, Y)$ coordinates of one point. A $k$–$d$ tree is a binary tree that enable fast query of nearby nodes. Therefore, we could quickly find all pairs of points within a certain distance. Any two points $(x_1, y_1)$ and $(x_2, y_2)$ were grouped into one cluster if $|x_1 - x_2| < G$ and $|y_1 - y_2| < G$. For each cluster, if the number of points in the cluster was more than a user-defined threshold (default: 5), it was considered as a potential region of enriched fragment endpoints. If the points in the cluster were not within a $G \times G$ square region, we used a $G \times G$ moving square to find a square region where the points are best enriched. Theoretically, the best enriched square region should contain 98.01% $(0.99 \times 0.99)$ of the points, according to Eq. (2). The predicted breakpoints were the $X$ and $Y$ coordinates of the right-bottom vertex of the square. The points in the square region were subjected to a statistical test describe below.

**Quantification of type 1 evidence**. Let $n$ be the number of points in the square region. Each point corresponds to a pair of fragment $F_{i1}$, $F_{i2}$, $(i = 1, 2, 3, \ldots, n)$ that may support the SV. Let $b_1$ and $b_2$ be the coordinates of the predicted breakpoint. Eqs. (1) and (2) hold for all the fragment pairs $F_{i1}$, $F_{i2}$ $(i = 1, 2, 3, \ldots, n)$. We then test the null hypothesis that there is no SV between $b_1$ and $b_2$.

First, we test the hypothesis that the $n$ fragment pairs $F_{i1}$, $F_{i2}$ have originated from different DNA molecules, but coincidently received the same barcode. Here, we define two fragments $F_a$ and $F_b$ as an independent fragment pair if $F_a$ and $F_b$ share the same barcode but have originated from different DNA molecules. Thus, $R(F_a)$ and $L(F_b)$ are independent variables. All the fragment pairs that do not support SVs are independent fragment pairs. It is reasonable to assume the generation of HMW DNA molecules from chromosomal DNA is a random process thus both $R(F_a)$ and $L(F_b)$ are uniformly distributed across the chromosome. Therefore, the point $((R(F_a), L(F_b))$ is equal likely to be in any place in the two-dimensional plane. Technically, we connect all the chromosomes in a head-to-tail order so that both intra-chromosomal events and interchromosomal can be analyzed at the same time. Observing at least $n$ independent fragment pairs meeting Eq. (2) is equivalent to the event that observing at least $n$ points $((R(F_{i1}), L(F_{i2}))$ located in a squared region with an area of $G^2$ on the two-dimensional plane. The probability of this event is

$$p_1 = \sum_{j=n}^{N} \text{Binomial\_pmf} \left( n, N_{\text{ifp}}, \frac{G^2}{L^2} \right), \tag{3}$$

where Binomial_pmf is the probability mass function of binomial distribution; $L$ is the total length of the genome (also the side length of the two-dimensional plane); $N_{\text{ifp}}$ is the total number of independent fragment pairs.

Since we are doing multiple hypothesis testing in the data set, the probability need to be adjusted.

$$p_{\text{adjusted1}} = p_1 \frac{G^2}{L^2}. \tag{4}$$

We reject the hypothesis if $p_{\text{adjusted1}} < p_{\text{threshold}}$. $p_{\text{threshold}}$ is $10^{-5}$ by default.

Next, we test the hypothesis that fragment pairs $F_{i1}$, $F_{i2}$ $(i = 1, 2, 3, \ldots, n)$ have originated from the same DNA molecule, but no reads were sequenced in the gap between $R(F_{i1})$ and $L(F_{i2})$. Let $g_i$ denote the length of the gap between $F_{i1}$ and $F_{i2}$, $\bar{g}$ denote the mean of $g_i$, and we have

$$g_i = L(F_{i2}) - R(F_{i1}), \tag{5}$$

$$\bar{g} = \frac{1}{n} \sum_{i=0}^{n} g_i. \tag{6}$$

If $\bar{g}$ is too large such that the probability of no reads being generated is smaller than a threshold, we can reject this hypothesis.

Similar to the model described by 10x Genomics[12], we assume the read generation on a DNA molecule is a Poisson process with constant rate $\lambda$ across the genome. Let $r$ be the number of reads generated in a region of length $g$, then $r \sim \text{Pois}(\lambda g)$. Let $P_{\text{gap}}(g)$ denote the probability of no read being generated in length $g$, we have

$$P_{\text{gap}}(g) = P(r = 0 | \lambda g) = \frac{e^{-\lambda g}(\lambda g)^0}{0!} = e^{-\lambda g}. \tag{7}$$

Therefore, the gap length $g_i$ follows exponential distribution: $g_i \sim \exp(\lambda)$. Recalling that (1) the exponential distribution with rate parameter $\lambda$ is a Gamma distribution

with shape parameter 1 and rate parameter $\lambda$; (2) the sum of $n$ independent random variables from Gamma $(1, \lambda)$ is a Gamma random variable from Gamma $(n, \lambda)$, we have

$$\sum_{i=0}^{n} g_i \sim \text{Gamma } (n, \lambda), \qquad (8)$$

$$\bar{g} = \frac{\sum_{i=0}^{n} g_i}{n} \sim \text{Gamma } (n, n\lambda). \qquad (9)$$

Therefore, the probability that observing $n$ gap regions with mean length equal to or larger than $\bar{g}$ is

$$p_2 = 1 - \text{Gamma\_cdf } (n, n\lambda). \qquad (10)$$

Where Gamma_cdf is the cumulative distribution function of Gamma distribution.

Since we are doing multiple hypothesis testing in the data set, the probability need to be adjusted.

$$p_{\text{adjusted2}} = p_2 \frac{N_{\text{rp}}}{n}, \qquad (11)$$

where $N_{\text{rp}}$ is the total number of read pairs.

We reject the hypothesis if $p_{\text{adjusted2}} < p_{\text{threshold}}$. $p_{\text{threshold}}$ is set as $10^{-5}$ by default. If both $p_{\text{adjusted1}}$ and $p_{\text{adjusted2}}$ are less than $p_{\text{threshold}}$, we accept the hypothesis that the SV is true. For each candidate SV, we report a confidence score for type 1 evidence as

$$\text{Confidence score 1} = -\log_{10}\Big(\max\big(p_{\text{adjusted1}}, p_{\text{adjusted2}}\big)\Big). \qquad (12)$$

**Breakpoint detection from type 2 evidence.** Barcode similarity between two nearby regions is very high because the reads originate from almost the same set of HMW DNA molecules. However, at the SV breakpoint, the aligned reads from the left side and right side may have originated from different locations in the alternative genome. Thus, the barcode similarity between the left side and right side of the breakpoint are dramatically reduced (as described in the Result section and shown in Fig. 1e, f). To detect this, LinkedSV uses two adjacent sliding windows (twin windows, moving 100 bp) to scan the genome and calculate the barcode similarity between the twin windows. The window length can be specified by user. By default, it is $G$ for WGS data sets and 40 kb for WES data sets.

The barcode similarity can be simply calculated as the fraction of shared barcodes. This method is suitable for WGS, where the coverage is continuous and uniform. But it does not perform well for WES, where the numbers of reads in the sliding windows vary a lot due to capture bias and the length of capture regions. Therefore, we use a model that considering the variation of sequencing depth and capture region positions. The barcode similarity is calculated as

$$S = \frac{x}{m_1^a m_2^b} n e^{-\alpha d}, \qquad (13)$$

where $m_1$ is the number of barcodes in window 1, $m_2$ is the number of barcodes in window 2, $x$ is the number of barcodes in both windows, $d$ is the weight distance between reads of the left window and the right window, $n$ is a constant representing the characteristic of the library, $\alpha$ is a parameter of fragment length distribution, a and b are two parameters between 0 and 1, $n$, $\alpha$, $a$, and $b$ are estimated from the data using regression. Detailed explanation of this model is in Supplementary Note 1.

Next, we calculate the empirical distribution of barcode similarity. Regions where the barcode similarity less than a threshold (5th percentile of the empirical distribution by default) were regarded as breakpoint candidates. If a set of consecutive regions have barcode similarity lower than the threshold, we only retain the region that has the lowest barcode similarity. If the barcode similarity of a breakpoint candidate is $S_0$, the empirical $p$ value is calculated as

$$p_{\text{empirical}} = \frac{\text{number of twin windows with } S \leq S_0}{\text{total number of twin windows}}. \qquad (14)$$

The confidence score of type 2 evidence is

$$\text{Confidence score 2} = -\log_{10}\big(p_{\text{empirical}}\big). \qquad (15)$$

**Combination of both types of evidence.** Type 1 evidence gives pairs of endpoints that indicate two genomic positions are joined in the alternative genome. Type 2 evidence gives genomic positions where the barcodes suddenly changed, regardless of which genomic position can be joined. Therefore, type 1 and type 2 evidence are independent. The candidate breakpoints detected from type 2 evidence were searched against the candidate breakpoint pairs detected from type 1 evidence so that the calls were merged. The combined confidence score is

$$\text{Combined score} = \text{Confidence score 1} + \text{Confidence score 2a} \\ + \text{Confidence score 2b}, \qquad (16)$$

where Confidence score 1 is the confidence score calculated from type 1 evidence (Eq. (12)); Confidence score 2a and Confidence score 2b are the confidence scores of the two breakpoints calculated from type 2 evidence (Eq. (14)).

**Refining breakpoints using short-read information.** For large SV events, we search for discordant read-pairs and clipped reads that are within 10 kb to the predicted breakpoint pairs by the above approach. We use a graph-based approach that is similar to DELLY[3] to cluster the discordant read-pairs. We define the supporting split-reads as the clipped reads that can be mapped to the both breakpoints, and the map direction matches the SV type. If both discordant read-pairs and split-reads are found to support the SV, we use the breakpoints inferred by split-reads as the final breakpoint position.

**Detection of small deletions that are within 50 bp–10 kb.** We use a 1 Mb moving window (with 0.1 Mb overlapping) to scan the genome. For each window, all the aligned reads (including phased and un-phased reads) were extracted and were assembled by the FermiKit pipeline. Regions with extreme high coverage (more than 20-fold of average coverage) were skipped. The resulting contigs were mapped back to the 1 Mb reference sequence of the moving window using bwa-mem and deletions were called from the aligned contigs if the alignments were unique within the 1 Mb moving window. The local assembly based process mainly contribute to the detection of deletions within 50–1000 bp. To detect deletions that are larger than 1 kb and might be missed by the assembly based process, we use a 500 bp moving window (with no overlapping) to find candidate regions where the read depth of either haplotype is less than 10% of the average depth of the haplotype. Next we extract all the read pairs of this haplotype and test if the mean insert size of these read pairs is significantly larger than the mean value of the whole genome, assuming the average insert size of $n$ read pairs follows normal distribution: $N(\mu, \sigma^2/n)$, where $\mu$ and $\sigma$ are the mean and standard deviation of the insert size of the whole genome.

We use a read depth based method to detect deletions that are larger than 1 kb and lack read pair support. If there are $m$ consecutive windows where the read depths are less than 10% of the average depth, we assume the read depth of each window is independent, and calculate the $p$ value using the simple equation: $p = (a/b)^m$, where $b$ is the total number moving windows and $a$ is the total number of moving windows where the read depths are less than 10% of the average depth. A deletion is called if $p < 10^{-10}$.

**Generation of simulated linked-read WGS data set.** The linked reads were simulated by LRSIM, which can generate linked-reads from a given FASTA file containing the genome sequences. We generated a diploid FASTA file based on hg19 reference genome with SNPs and SVs inserted. The purpose of inserting SNPs was to mimic real data. The generation of the diploid FASTA file is described below. First, we inserted SNPs to hg19 using vcf2diploid[31]. The inserted SNPs were from the gold standard SNP call set (v.3.3.2) of NA12878 genome[32]. The vcf2diploid software generated two FASTA files, each of which was a pseudohaplotype (paternal or maternal) with the phased SNPs inserted. Next, we insert SVs into the paternal FASTA file using our custom script. The breakpoints were located in the repetitive regions in hg19 and the distance between the two breakpoints were in the range of 50 kb to 10 Mb. In total, we simulated 351 deletions, 386 duplications, 353 inversions and 85 translocations, all of which were in the paternal copy and were heterozygous SVs. We then concatenate the paternal and maternal FAST file into a single FASTA file and simulated linked-reads using LRSIM. To mimic real data, the barcode sequences and molecule length distribution used for simulation were from the NA12878 whole-genome data set released by 10x Genomics. The number of read pairs was set to 360 million so that a 35× coverage data set was generated. The genome coordinates of simulated SVs was shown in Supplementary Data 1. The size distribution of the simulated SVs was shown in Supplementary Fig. 16a.

**Generation of WGS data set with low VAF.** In cancer samples or mosaic samples, the total DNA is a mix of a small portion of variant alleles and a large portion of normal alleles. To simulate the WGS data sets with low variant frequencies, we used the same paternal and maternal FASTA file described above but the combined FASTA file contained multiple copies of the normal allele (the maternal FASTA) and only one copy of the variant allele (the paternal FASTA). For example, to simulate a WGS data set with VAF of 20%, four copies of the maternal FASTA and one copy of the paternal FASTA were combined. The linked reads were simulated using LRSIM with the same parameters and a 35× coverage data set was generated.

**Simulation of deletions and duplications that cause diseases.** To test the performance of LinkedSV on the detection of disease casual SVs, we downloaded a list of expert-curated deletions and duplications that were known to cause CNV syndromes involved in developmental disorders. This list was downloaded from the DECIPHER database, and contained 67 CNV syndromes. Some syndromes were affected by CNV events in the same region. After removing redundant syndromes, we got 51 CNV events (Supplementary Table 10). Based on the 51 CNV events we simulated a germline WGS data set and two mosaic WGS data sets (VAF = 10% and 20%) using the same method described above.

**Generation of simulated linked-read WES data set.** To generate the linked-read WES data set, we first generate a 100× linked-read WGS data set and then down-sample it to be a WES data set. Generation of the simulated linked-read WGS data set with SNPs and SVs inserted was similar to the method described above. In total,

we inserted 1160 heterozygous SVs. The SV breakpoints were randomly selected from regions that were within 2000 bp of an exon. Among the 2320 breakpoints (two breakpoints per each SV), 1028 breakpoints (44.3%) were in intronic or intergenic regions. The SV sizes are in the range of 50 kb to 10 Mb (Supplementary Fig. 16b). Supplementary Data 2 showed the list of simulated SVs. The number of inserted SVs in the simulated WES data set was slightly smaller than that in the simulated WGS data set because the SV breakpoints were designed to reside within 2000 bp of an exon. The simulated reads were generated using LRSIM and were mapped to hg19 reference genome using the Longranger pipeline (default settings). The phased bam generated by Longranger was down-sampled to be a simulated WES data set. To mimic real WES data set, we used the coverage distribution of the linked-read WES data set of NA12878 genome (released by 10x Genomics) to guide the down-sampling process. We bin the genome into 10 bp windows and calculate number of reads mapped to each window (left mapping positions were used) in NA12878 linked-read WES data. The simulated WES data set was generated by sampling reads from the 100× WGS data according to number of reads mapped to the same 10 bp window in the NA12878 WES. The down sampling was at read pair level, if the one read is retained, the paired read would also be retained.

**Benchmarking of deletion detection on the HG002 genome**. The HG002 benchmark set (version 0.6) was downloaded from the FTP site: ftp://ftp-trace.ncbi. nlm.nih.gov/giab/ftp/data/AshkenazimTrio/analysis/NIST_SVs_Integration_v0.6/. The benchmarking process was performed according to the authors' suggestions[21]. The benchmark set contains a Tier 1 benchmark regions, where all the insertions/ deletions are resolved and any extra calls were putative false positives. This region covers 2.66 Gbp of the human genome. A deletion call was considered to be a true positive call if it had at least 50% reciprocal overlap (the overlapped region was more than 50% of both calls) with a deletion call with filter = PASS in the Tier 1 vcf file. Otherwise, it was considered to be a false-positive call. This 50% reciprocal overlapping criterion was chosen to follow what was done by a previous study[33].

Recall, precision and F1 score were calculated as follows.

$$\text{Recall} = \frac{\text{Number of true positive calls}}{\text{Total number of deletion calls with filter} = \text{PASS in the Tier 1 vcf file}}. \quad (17)$$

$$\text{Precision} = \frac{\text{Number of true positive calls}}{\text{Total number of deletion calls of the query set}}. \quad (18)$$

$$\text{F1 score} = \frac{2 * \text{Recall} * \text{Precision}}{\text{Recall} + \text{Precision}}. \quad (19)$$

**Reporting summary**. Further information on research design is available in the Nature Research Reporting Summary linked to this article.

## Data availability

The 10x Genomics sequencing data of the HX1 genome was generated in this study and can be obtained from the NCBI SRA database with the accession code SRX5781869.

The PacBio sequencing data of the HX1 genome was previously published and can be obtained from the NCBI SRA database with the accession code SRX1424851. The 10x Genomics sequencing data of the HG002 genome was released by 10x Genomics and can be downloaded from https://support.10xgenomics.com/de-novo-assembly/datasets/2.1.0/ash.

Due to potential compromise of individual privacy, full datasets of the clinical samples (*F8* and *NF1*) are available from the authors on reasonable request and institutional data use agreement. All other relevant data is available upon request.

## Code availability

The source code of LinkedSV is publicly available on GitHub (https://github.com/WGLab/LinkedSV). A detailed description of how to use LinkedSV is also provided in the GitHub repository.

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

## Acknowledgements

The authors would like to thank members of the Center for Applied Genomics for generating the linked read exome sequencing data on the patient with *F8* inversion and *NF1* deletion. We would like to thank the authors of the simulation software LRSIM to provide tools that facilitated our benchmarking study. This study is in part supported by NIH grant GM132713 to K. Wang.

## Author contributions

L.F. and K. Wang designed the study. L.F. implemented the tool and performed the analysis. F.A.M. and R.P.S. generated the 10x Genomics sequencing data of the *F8* inversion sample and the *NF1* deletion sample and C.K. and M.V.G analyzed the data. S.W. and K. Wimmer performed targeted Illumina MiSeq sequencing of the *NF1* deletion sample and analyzed the data. M.L. and H.H. guided on the method development and data analysis. L.F. drafted the paper. All authors read, revised, and approved the paper.

## Competing interests

The authors declare no competing interests.
