## [Peer Review File · Nature Communications]

Reviewers' comments:

Reviewer #1 (Remarks to the Author):

Review: LinkedSV: Detection of mosaic structural variants from linked-read exome and genome sequencing data

The manuscript by Fang et al. proposes a novel algorithm SV caller for linked-read sequencing using two approaches to identifying SV signatures or "evidence". One type of evidence identifying which genomic positions are connected in the alternative genome and the second type looking at which genomic position is interrupted with differing barcodes on the left and right of the breakpoint. They demonstrate the novelty and performance of their method using three simulated data sets and two real data sets.

The authors utilize comparisons with three other tools, namely Longranger, GROC-SV, and NAIBR to highlight their contributions to this space and make admirable efforts of detailing the differences between LinkedSV and these publicly available SV callers. They also show the improvement to be gained by using linked-reads for SV calling not only over short-read sequencing but also long read sequencing. They highlight the ability of LinkedSV to achieve higher recall than other tools but also its ability to identify low VAF SVs and also perform relatively well on whole exome sequencing data.

Overall, the authors do a good job of breaking down their algorithm and methodology in the development of LinkedSV. However, I believe there are still a few things missing from the manuscript in order to complete the study. As is standard with these studies, distribution of the size of the events called by LinkedSV is necessary to give a more accurate picture of the capabilities of the tools. The authors mention that SVs simulated were between 50 kb to 1 Mb but it would be useful to see what are the sizes of events called when LinkedSV is used on real data and how do these size boundaries change for each SV type. This discussion should

Also, is a user-defined threshold for what constitutes an enriched cluster a better approach than enriched being based on what is expected at an average position across the genome in terms of fragment endpoints? The authors also discuss in their algorithm the idea of a moving GxG square to capture "most of the points" but give no information as to what that realistically means. Is that a user-defined parameter? What is currently being used in the results as the cutoff for most, >50%?

It's understood that regions of segmental duplication can be challenging but the authors state that SMRT-SV was able to find 194 large duplications that LinkedSV did not use long read data. But did not necessarily explain why SMRT-SV were able to catch these especially considering that both long-read (which SMRT-SV uses) and linked reads have low map quality in the regions as they mentioned.

The authors include multiple statistical tests to attempt to recover the most confident of calls which is also commendable and discuss how they are utilizing essentially two approaches to SV calling where most callers use one. With this in mind, however, the manuscript is lacking in exploring the computation time of linkedSV when compared to currently used tools, especially on real data.

On a more minor note, I'd recommend proofreading the manuscript again to catch a few grammatical errors. For example, this is an excerpt from the ms "We now describe how we use to detect the type 1 evidence". Also, citations are necessary when making statements such as these, "large SVs are more likely to be harmful and to cause diseases."

Overall, I do think this study will advance the linked-read field. I have a few additional concerns that need to be addressed as well:

- "To generate the linked-read WES data set, we first generate a 100X linked-read WGS data set and then down-sample it to be a WES data set". I believe that experiment design is not accurate. Here we set aside all possible spliced reads and because of that, we cannot see paired read signal at SV breakpoints. This may lead to results deterioration for some methods (e.g. GROC-SV requires paired-end signal when calling SVs) and wrong conclusions about GROC-SV and short read SV callers' performance. So, Figure 3a should be redone.

- a novel SV detection algorithm which combines two types of statistical evidence - Uninformative, better to describe what does it mean or leave it or just write "statistical framework"

- "A probabilistic approach is used to clean up this initial candidate list" – It is not really clear what the authors mean here.

- "which is typically longer than 5 kb for whole-genome sequencing (WGS) data sets" - Here you need some references. I don't think this statement is necessarily correct. Basically, I understand why this is mentioned. If deletion is short, it is hard to distinguish between short deletion or no SV cases. You can also mention the shortest possible SV you aim to detect. Again it is important to emphasize the range of event sizes you can detect.

- Figure 1b requires better legend and caption. probably showing the reference genome or transformation pattern on the same figure would help.

- barcode similarity between the two nearby window regions are significantly decreased." -> is decreased

- "The inserted SVs includes 351 deletions, 386 duplications, 353 inversions, and 85 translocations." - include into supplementary?

- From the Figures 2b-c it is not clear why authors think that this is an example of a duplication event. Figure 2c doesn't help to support this claim.

- "44.3% of the breakpoints are 207 not in exon regions" – probably better to move this sentence to Methods?

- In sections "Performance evaluation on simulated whole-exome sequencing data " and "Detection of F8 inversion from clinical WES data" there is a deletion/insertion that lacks paired-end read support. However, I don't understand how it is possible. While it is clear that we can use shared barcode, in my opinion, some read pair should have discordant mappings. Is it a mapper limitation?

- "We assign G as the 99th percentile of the empirical distribution of intra-fragment gap distance" - From the text, it seems that each fragment should be split into two fragments. I suppose that it is not true, need a more accurate description

- What are the limitations of your method? Maybe a better discussion would be helpful.

Reviewer #2 (Remarks to the Author):

The authors of the manuscript "LinkedSV: Detection of mosaic structural variants from linked-read 2 exome and genome sequencing data" describe a new method to detect Structural Variations using linked reads from 10x genomics. Linked reads have the potential to help in the detection of SVs as they have an improved mapability also in highly repetitive regions. Exactly those regions that often include structural variations. Thus, it is important to further develop methods as the currently default method (using Longranger) often fails to detect SVs. Nevertheless, your comparisons and benchmarks are lacking sometimes of details and precision.

The manuscript is well written and easy to follow. In the following I list my questions and concerns.

1. Why did you restrain the evaluation of 5kbp/10kbp or larger events? A lot of other SVs impacting certain phenotypes have been reported that are much shorter.
2. I am wondering why you could not identify clipped reads by the HX1 PacBio reads. Can you rule out that this might be an artifact in either technologies?
3. For your simulation you only simulated events between 50k to 1Mbp. This is of course very biased towards the linked reads. How does your method perform on smaller events of a few 100bp?
4. I did not understand why you did the error correction for the PacBio reads and which of these reads were used where in the evaluation. The error correction can potentially erase some of the deletion events, but I agree that this should happen more often for the smaller events rather than these large events.
5. I noticed in the mapping parameter for Minimap2 you mention you are using the Nanopore preset parameter for the PacBio reads. Please check that!
6. Its not clear what parameters you used for other tools such as Delly, Lumpy, Sniffles etc. Also why didn't you use Delly, Lumpy across the study?
7. Maybe I missed it but in the discussion you mention the detection of novel insertions (line 587). However, I did not see that benchmark.
8. In your discussion you also mention that you showed that the sequencing error of long reads negatively impacts the read mapping. I did not see this comparison.
9. You mention that SNPs and indels can be detected from linked reads, but this is also true for all other technologies.
10. Your claim that large SV are more likely to impact a phenotype is not supported. If you have proof, please provide citations especially since you use it as a motivation to justify that your method was not benchmarked below 10kbp SV length.
11. I would also recommend using the gold standard data set from GIAB HG002 which also includes high quality 10x genomics data.

Reviewer #3 (Remarks to the Author):

The authors present LinkedSV, a new method for detecting structural variants in 10X data. This method uses the similarity between the sets of bar codes in different regions to help identify variants.

The paper is well written, the process is interesting, and the performance of LinkedSV was better than other methods. I also liked the section about detection in WES, which is an area that needs better SV detection methods.

My major issue with this paper is its reliance on simulated data. While I am very sympathetic to the motivations of using simulations, especially considering that there are no good truth sets for SVs, it is difficult to validate the simulation itself. The authors biased their simulations toward repetitive regions based on the observation that these regions are more likely to harbor SVs. On the surface this seems like a good idea, but how faithful is the simulation to this biologic observation? To me, it seems more reasonable to simulate 10X over known SVs in these repetitive regions, instead of randomly picking something.

I am also troubled by the differences between the germline and the cancer simulation. In particular, that GROCSV went from matching LinkedSV's performance in one simulation to detecting nothing in the next, while the NAIBR went from being one of the worst methods to the only other method to detect anything. I understand that the allele balance shifts between these two, but detecting nothing makes me think something is wrong with the simulation or the way the tools were run. Other SV methods (not 10X methods) have done similar experiments, and I have never seen a report of a high-quality caller that "almost completely failed."

In the section that compared long-reads to linked-reads, I would like the authors to dig in a bit more as to why the SVs that were not detected using long reads but picked up by LinkedSV are not false positives (beyond visual inspection). For example, the chr19 duplication spans a region that is densely tiled with microsatellites, which are a well known source of false positives for short read alignment. Maybe the reason that long reads do not show extra coverage here is that the long reads are correctly mapped whereas the short reads are not.

I am sure that LinkedSV does better than other SV callers that do not consider the bar codes, but it would be interesting to see if there is any reduction in sensitivity when you start by looking for the barcode pattern.

Response to Reviewer's Comments

Summary of changes

We would like to thank the editor and the reviewers for the evaluation of our manuscript titled 'LinkedSV: Detection of mosaic structural variants from linked-read exome and genome sequencing data'. We have carefully addressed all of the comments raised by the editor and the reviewers, and made corresponding modifications to the manuscript. In particular, we have substantially improved LinkedSV, so that it now incorporates read depth and paired-end signals, and it now uses a local assembly-based method for small deletions to increase sensitivity. Furthermore, we have performed additional benchmarking studies on real and simulated data sets to evaluate the performance under various scenarios for different types and sizes of SVs. By addressing the reviewers' comments, we believe that the qualities of the manuscript and the software tool are greatly improved.

The point to point responses to reviewers' comments are given below:

Responses to reviewer 1

The manuscript by Fang et al. proposes a novel algorithm SV caller for linked-read sequencing using two approaches to identifying SV signatures or "evidence". One type of evidence identifying which genomic positions are connected in the alternative genome and the second type looking at which genomic position is interrupted with differing barcodes on the left and right of the breakpoint. They demonstrate the novelty and performance of their method using three simulated data sets and two real data sets.

The authors utilize comparisons with three other tools, namely Longranger, GROC-SV, and NAIBR to highlight their contributions to this space and make admirable efforts of detailing the differences between LinkedSV and these publicly available SV callers. They also show the improvement to be gained by using linked-reads for SV calling not only over short-read

sequencing but also long read sequencing. They highlight the ability of LinkedSV to achieve higher recall than other tools but also its ability to identify low VAF SVs and also perform relatively well on whole exome sequencing data.

Overall, the authors do a good job of breaking down their algorithm and methodology in the development of LinkedSV. However, I believe there are still a few things missing from the manuscript in order to complete the study.

Response: Thank you for the nice summary of the manuscript and the tool. Our point by point responses are given below.

Reviewer comments: *As is standard with these studies, distribution of the size of the events called by LinkedSV is necessary to give a more accurate picture of the capabilities of the tools. The authors mention that SVs simulated were between 50 kb to 1 Mb but it would be useful to see what are the sizes of events called when LinkedSV is used on real data and how do these size boundaries change for each SV type. This discussion should*

Response: Thank you for the comment. Based on reviewer 2's suggestions, we benchmarked LinkedSV's performance on the HG002 genome. The sizes of events called by LinkedSV on the HG002 genome were shown in Supplemental Figure 15 (page 77, line 1147). Since the barcode information provides little information on resolving small SVs (< 10 kb), the original version of LinkedSV only aimed at the detection of large SVs. During the revision process, to fully address several reviewers' comments, we made improvements to the software tool and added the feature of detection of small deletions by using multiple sources of information, including read depth, paired-end reads and local assembly of the short reads. Therefore, the updated LinkedSV, as shown in the revised version, is able to detect small deletions up to 50 bp. By addressing these comments, we believe that the quality of the tool is improved significantly.

The sizes of simulated SVs were shown in Supplemental Figure 8 (page 69, line 1086). Please note that the simulated SVs were between 50 kb to 10 Mb, but not 1 Mb: we have corrected this

typo in the revised manuscript and apologize for the typo. However, only a small portion of the simulated SVs are > 1Mb. 72% (WGS data set) and 83% (WES data set) of the simulated SVs were within 50 kb to 1Mb.

Reviewer comments: *Also, is a user-defined threshold for what constitutes an enriched cluster a better approach than enriched being based on what is expected at an average position across the genome in terms of fragment endpoints? The authors also discuss in their algorithm the idea of a moving $G \times G$ square to capture “most of the points” but give no information as to what that realistically means. Is that a user-defined parameter? What is currently being used in the results as the cutoff for most, >50%?*

Response: Thank you for the comment. Advanced users can use the “--gap_distance_cut_off” parameter to specify the threshold (i.e. the value of G). For general users, we calculate the value of G according to the statistics from the data. The detailed method for calculating G was described in the Method section (page 27, lines 481-487). Briefly, G is the 99th percentile of the empirical distribution of gap distance (i.e. the distance of two nearby reads in the same long fragment). Theoretically, almost all ($99\% \times 99\%$) of the points should be within a $G \times G$ square. We have added this description in the Method section (pages 28-29, lines 511-517). The goal of using $G \times G$ moving square is to find the square region where the points are enriched the most. We didn’t set a hard cutoff of how many points should be within this square region but theoretically the best enriched one should contain 98.01% of the points that supporting an SV. We have added this description in lines 534-536 in page 30. For a typical 30X WGS data set, G is about 10 kb.

Reviewer comments: *It’s understood that regions of segmental duplication can be challenging but the authors state that SMRT-SV was able to find 194 large duplications that LinkedSV did not use long read data. But did not necessarily explain why SMRT-SV were able to catch these especially considering that both long-read (which SMRT-SV uses) and linked reads have low map quality in the regions as they mentioned.*

Response: Thank you for pointing this out. SMRT-SV was able to detect large duplication although both long reads and linked reads have low map quality in these regions. One possible reason is that, SMRT-SV doesn't rely on the alignment of PacBio raw reads. SMRT-SV detects SVs using an assembly-based approach. For reads generated by the PacBio RS II system, SMRT-SV first used Celera assembler to get the local assemblies of the long reads, and map the contigs back to the reference genome using BLASR. SVs were called from the alignments of contigs. During the assembly process, the assembly contigs were error corrected and polished by the PacBio reads. Therefore, the assembly contigs are potentially more accurate and longer than each of the raw reads. Thus, it is possible for SMRT-SV to detect SVs in these large segmental duplication regions. We added additional discussions on SMRT-SV in the revised manuscript to clarify this point (page 25, lines 449-455).

***Reviewer comments:** The authors include multiple statistical tests to attempt to recover the most confident of calls which is also commendable and discuss how they are utilizing essentially two approaches to SV calling where most callers use one. With this in mind, however, the manuscript is lacking in exploring the computation time of linkedSV when compared to currently used tools, especially on real data.*

Response: Thank you for pointing this out. We have added the comparison of computation time on the 37X HX1 whole-genome sequencing data set. Please refer to Supplemental Figure 12 (page 75, line 1131). The computation time of the new version of LinkedSV is longer than NAIBR because it uses multiple types of evidence and also it now performs local assembly to detect small deletion events.

***Reviewer comments:** On a more minor note, I'd recommend proofreading the manuscript again to catch a few grammatical errors. For example, this is an excerpt from the ms "We now describe how we use to detect the type 1 evidence".*

Response: Thank you for pointing this out. We have corrected this sentence in page 23, line 407.

Reviewer comments: Also, citations are necessary when making statements such as these, “large SVs are more likely to be harmful and to cause diseases.”

Response: We have changed this sentence to “according to our analysis of SV size distribution (Supplementary Note 2), large SVs are associated with diseases such as cancers and CNV syndromes”. Intuitively, larger SVs have a higher chance to span a coding region or a functional element in the genome than smaller SVs. In addition, we analyzed the SV size distribution from two resources including 1) COSMIC somatic cancer mutation database, 2) a list of expert-curated CNV syndromes from DECIPHER database. The results showed that 71% of the somatic cancer SVs in COSMIC database are larger than 10 kb and all deletions/duplications that cause the CNV syndromes are larger than 10 kb.

Reviewer comments: Overall, I do think this study will advance the linked-read field. I have a few additional concerns that need to be addressed as well:

“To generate the linked-read WES data set, we first generate a 100X linked-read WGS data set and then down-sample it to be a WES data set”. I believe that experiment design is not accurate. Here we set aside all possible spliced reads and because of that, we cannot see paired read signal at SV breakpoints. This may lead to results deterioration for some methods (e.g. GROC-SV requires paired-end signal when calling SVs) and wrong conclusions about GROC-SV and short read SV callers’ performance. So, Figure 3a should be redone.

Response: Thank you for the comment. As described in the Method section (line 729 of page 41), the down-sampling was at read-pair level. Read-1 and read-2 were retained or discarded at the same time. In our implementation, if the one read pair was retained, all the alignments (including primary, secondary, supplementary alignments) of both read-1 and read-2 would be retained in the bam file. Therefore, paired-read signals are retained and can be used by GROC-SVs, NAIBR, and also LinkedSV.

According to reviewer 2's suggestions, we evaluated the performance of Delly and Lumpy in the simulated data sets as well. Delly and Lumpy are short-read SV callers which use paired-end signal. As shown in the updated Figure 4a (page 56), Delly and Lumpy detected 56.9% and 51% of the simulated SV calls, respectively. This indicates that the paired-end signals are indeed retained in the simulated WES data set.

Reviewer comments: *a novel SV detection algorithm which combines two types of statistical evidence - Uninformative, better to describe what does it mean or leave it or just write "statistical framework"*

Response: Thank you for the comment. We have changed the description in the abstract (page 2, line 21).

Reviewer comments: *"A probabilistic approach is used to clean up this initial candidate list" – It is not really clear what the authors mean here.*

Response: Thank you for the comment. We have changed the description in the manuscript (page 4, lines 69-70).

Reviewer comments: *"which is typically longer than 5 kb for whole-genome sequencing (WGS) data sets" - Here you need some references. I don't think this statement is necessarily correct. Basically, I understand why this is mentioned. If deletion is short, it is hard to distinguish between short deletion or no SV cases. You can also mention the shortest possible SV you aim to detect. Again it is important to emphasize the range of event sizes you can detect.*

Response: Thank you for pointing this out. Here the "5kb" means the typical size of deletions that can be detected by this type of evidence, not the typical size of deletions in human individuals. We have changed the description to be "This can be observed in deletions with minimal size of about 5-10 kb" (page 7, line 110).

Reviewer comments: *Figure 1b requires better legend and caption. Probably showing the reference genome or transformation pattern on the same figure would help.*

Response: Thank you for pointing this out. The detailed explanations of how the patterns are formed can be found in Supplemental Figure 1-3. We have added this information in the legend of Figure 1b.

Reviewer comments: *barcode similarity between the two nearby window regions are significantly decreased.” -> is decreased*

Response: Thank you for pointing this out. We have corrected this in the manuscript (page 8, line 134).

Reviewer comments: *“The inserted SVs includes 351 deletions, 386 duplications, 353 inversions, and 85 translocations. ” - include into supplementary?*

Response: Thank you for the comment. We have removed this sentence in the Result section because this was also described in the Method section (page 38, line 684). The genome coordinates of the simulated SVs were shown in Supplemental Tables 7.

Reviewer comments: *From the Figures 2b-c it is not clear why authors think that this is an example of a duplication event. Figure 2c doesn't help to support this claim.*

Response: Thank you for the comment. We have added an additional panel (figure 2d) showing the read depth around the region. From Figure 2d we can see that the increase of depth in the duplication region. Figure 2c shows the fragments that span the junction of the first copy and the second copy. Supplemental Figure 1 explained the pattern of enriched fragment endpoints for

tandem duplications. In addition, to help readers understand this pattern, we prepared a video showing the read alignment process and how this pattern is formed (Supplemental Movie 1).

Reviewer comments: *“44.3% of the breakpoints are not in exon regions” – probably better to move this sentence to Methods?*

Response: Thank you for the comment. We write this sentence here to emphasize that we can detect SVs from WES data even when the breakpoints are not in the exonic regions (capture regions). This is because we use the long-range information from the barcodes in linked-read sequencing, although we can additionally use paired-end read information to further identify the precise breakpoint locations.

Reviewer comments: *In sections “Performance evaluation on simulated whole-exome sequencing data ” and “Detection of F8 inversion from clinical WES data” there is a deletion/insertion that lacks paired-end read support. However, I don’t understand how it is possible. While it is clear that we can use shared barcode, in my opinion, some read pair should have discordant mappings. Is it a mapper limitation?*

Response: Thank you for the comment. In these two examples, the deletion/inversion breakpoints were not in exon regions/capture regions. Therefore, the read pairs with discordant mappings (which span the breakpoints) were not captured and were not in the data. However, with barcode information of the reads mapped to exons near the true breakpoints, we can predict the event. The predicted breakpoint locations are the nearest exons of the real breakpoint.

Reviewer comments: *“We assign G as the 99th percentile of the empirical distribution of intra-fragment gap distance” - From the text, it seems that each fragment should be split into two fragments. I suppose that it is not true, need a more accurate description.*

Response: Thank you for the comment. We didn't split every fragment. G was not a variable number depending on each fragment. The empirical distribution of intra-fragment gap distance was calculated from all the fragments (i.e. combine the intra-fragment gap distances of all fragments), and 99th percentile of this distribution was assigned to G . Therefore, G was a fixed number for all fragments. Only the fragments that potentially have SVs will be split. We apologize for the confusion, and we have changed the description in the manuscript to clarify this point (page 27, lines 485-487).

Reviewer comments: *What are the limitations of your method? Maybe a better discussion would be helpful.*

Response: Thank you for the comment. We have now discussed limitations in the Discussion section of the revised manuscript (page 25, lines 456-459). For example, LinkedSV has limited power to detect small duplications and inversions. Another limitation is that LinkedSV currently cannot detect insertions and repeat expansions (page 26, line 462).

Responses to Reviewer 2

The authors of the manuscript “LinkedSV: Detection of mosaic structural variants from linked-read 2 exome and genome sequencing data” describe a new method to detect Structural Variations using linked reads from 10x genomics. Linked reads have the potential to help in the detection of SVs as they have an improved mapability also in highly repetitive regions. Exactly those regions that often include structural variations. Thus, it is important to further develop methods as the currently default method (using Longranger) often falls to detect SVs. Nevertheless, your comparisons and benchmarks are lacking sometimes of details and precision.

The manuscript is well written and easy to follow. In the following I list my questions and concerns.

Response: Thank you for the nice summary of the manuscript and the tool. Our point by point responses are given below.

***Reviewer Comments:** Why did you restrain the evaluation of 5kbp/10kbp or larger events? A lot of other SVs impacting certain phenotypes have been reported that are much shorter.*

Response: Thank you for pointing this out. As we discussed in the Discussion section (page 43, lines 8-17), the linked-read technology provides strong evidence to detect large SVs, but it provides little additional evidence to detect small SVs. Therefore, LinkedSV has limited power to detect small SVs. During the revision process, we added the feature to detect small deletions by using multiple information including read depth, paired-end reads and local assembly of the short reads. The benchmarking of deletion detection on the HG002 genome was shown in Supplemental Figure 14.

Currently, LinkedSV is able to detect deletions ≥ 50 bp, inversions ≥ 10 kb, tandem duplications ≥ 20 kb and intra-chromosomal translocations of any size. We have described the sizes of SVs that can be detected in Supplementary Note 2 (page 91). In addition, we analyzed

the size of disease associated SVs from two resources including 1) COSMIC somatic cancer mutation database, 2) a list of expert-curated CNV syndromes from DECIPHER database. The results showed that 71% of the somatic cancer SVs in COSMIC database are larger than 10 kb and all deletions/duplications that cause the CNV syndromes are larger than 10 kb. Therefore, LinkedSV has strong potential to detect these types of SVs.

Reviewer Comments: *I am wondering why you could not identify clipped reads by the HX1 PacBio reads. Can you rule out that this might be an artifact in either technologies?*

Response: Thank you for the comments. Initially, when we examine the clipped reads, we only look at the reads with mapping quality > 20. After examine the deletion region in UCSC genome browser, we found that this 45 kb deletion reside in a 358 kb segmental duplication region (Supplemental Figure 9a, page 70). All the clipped reads had low mapping qualities. After examine all the aligned PacBio reads, we were able to find the clipped reads (Figure 7b-c). Further analysis of the clipped reads showed that this was a complex SV event. The 45 kb region in chr2 was deleted and a 6 kb sequence from the hs38d1 decoy sequence was inserted. As described in the manuscript (pages 19-20, lines 343-356), we aligned all the PacBio reads to a new reference genome with all sequences of GRCh38 plus hs38d1 and the sequence of the proposed variant allele. There were 33 reads spanning the chr2-hs38d1 junction, 48 reads spanning the hs38d1-chr2 junction and 13 reads spanning both junctions. *De novo* assembly of the reads aligned to the proposed variant allele generated a single contig of 42,715 bp, which also spanned both junctions (Supplemental Figure 9b, page 70). These analysis showed that the large deletion event was detected from both platforms and with PacBio long reads the details of complex SV events could be resolved.

Reviewer Comments: *For your simulation you only simulated events between 50k to 1Mbp. This is of course very biased towards the linked reads. How does your method perform on smaller events of a few 100bp?*

Response: Thank you for pointing this out. As we responded in the above comment, the linked-

read technology provides strong evidence to detect large SVs, but it provides little additional evidence to detect small SVs. Therefore, the original version of LinkedSV cannot detect SVs that are smaller than 10 kb. During the revision process, we added the feature of detection of small deletions by using multiple information including read depth, paired-end reads and local assembly of the short reads. The benchmarking of deletion detection on the HG002 genome was shown in Supplemental Figure 14.

***Reviewer Comments:** I did not understand why you did the error correction for the PacBio reads and which of these reads were used where in the evaluation. The error correction can potentially erase some of the deletion events, but I agree that this should happen more often for the smaller events rather than these large events.*

Response: Thank you for the comments. The motivation of using Canu to do error correction is to see if the error correction improves the alignment of long reads. As the long reads have an average error rate of about 15%, the aligners have a high tolerance to alignment mismatches and may prefer to generate an alignment with more mismatches/indels rather than clip it and align the clipped reads to another region.

After manual inspection of the duplication call of the HX1 genome (page 20, lines 364-368), a small duplication event was found next to the main event (Supplemental Figure 11, page 73). The boundaries of the small duplication can be observed in the alignments of linked reads and error-corrected PacBio reads, but not in the alignments of PacBio raw reads. There are enriched alignment mismatch in the red box of Supplemental Figure 11b, indicating that this portion of reads should be clipped, rather than aligned with a high mismatch rate.

Error correction has been used in some SV callers. For example, SMRT-SV uses Canu to assembly the reads and Canu internally performs error correction before the assembly step. As you mentioned, error correction may erase small insertion/deletion events, but it should not affect large events.

Reviewer Comments: *I noticed in the mapping parameter for Minimap2 you mention you are using the Nanopore preset parameter for the PacBio reads. Please check that!*

Response: Thank you for very much for pointing this out. We have checked our command and found that we used the correct parameter for the PacBio reads (-x ont-pb). We apologize for this typo.

Reviewer Comments: *It's not clear what parameters you used for other tools such as Delly, Lumpy, Sniffles etc. Also why didn't you use Delly, Lumpy across the study?*

Response: Thank you very much for this suggestion. We have added the benchmarking of Delly and Lumpy in all data sets (Figure 2a, Figure 3a and b, Figure 4a, Supplemental Figure 14, and Supplemental Figure 16). This greatly help the understanding how the linked-read SV callers perform, comparing to the conventional short read SV callers. We added how we use all the SV callers in the Method section (page 42, lines 745-758). We used the default parameters for Delly, and used the smooove pipeline to run Lumpy and filter the results as per the authors' suggestion. While we detect SVs from the HX1 PacBio data set using Sniffles, we used the "--min_support 1" parameter to set the minimum number of reads that support a SV to be 1. The purpose is to maximize the sensitivity and see if Sniffles can detected the duplications reported by SMRT-SV.

Reviewer Comments: *Maybe I missed it but in the discussion you mention the detection of novel insertions (line 587). However, I did not see that benchmark.*

Response: Thank you for pointing this out. As we described in the discussion section (page 26, line 461-463), LinkedSV currently does not handle insertions. As a future direction, we plan to detect novel sequence insertions using type 2 evidence.

Reviewer Comments: *In your discussion you also mention that you showed that the sequencing error of long reads negatively impacts the read mapping. I did not see this comparison.*

Response: Thank you for this comment. In Supplemental Figure 11, we compared the mapping of PacBio raw reads, PacBio error-corrected reads and linked reads at a duplication breakpoint in the HX1 genome. A small duplication event was found next to the main event. The boundaries of the small duplication can be observed in the alignments of linked reads and error-corrected PacBio reads, but not in the alignments of PacBio raw reads. There are enriched alignment mismatch in the red box of Supplemental Figure 11b, indicating that this portion of reads should be clipped, rather than aligned with a high mismatch rate.

***Reviewer Comments:** You mention that SNPs and indels can be detected from linked reads, but this is also true for all other technologies.*

Response: Thank you for pointing this out. In this paragraph, we were comparing the linked-read sequencing and long-read sequencing. There are a few tools such as DeepVariant that can detect SNPs and indels from long-read sequencing. However, detection of SNPs and indels from long reads is in its early stage and short-read sequencing is still the most widely used sequencing platform in clinical labs for SNP and indel detection. Linked-read sequencing is still relying on short-read sequencing but it provides long-range information by a special barcoding procedure during library preparation, so that clinical labs can use it to detect SNP, indels and SVs simultaneously.

***Reviewer Comments:** Your claim that large SV are more likely to impact a phenotype is not supported. If you have proof, please provide citations especially since you use it as a motivation to justify that your method was not benchmarked below 10kbp SV length.*

Response: Thank you for the comments. We have changed this sentence to “based on our analysis of SV size distribution, large SVs are associated with diseases such as cancers and CNV syndromes (Supplementary Note 2)”. Intuitively, larger SVs have a higher chance to span more

functional regions in the genome than smaller SVs. In addition, we analyzed the SV size distribution from two resources including 1) COSMIC somatic cancer mutation database, 2) a list of expert-curated CNV syndromes from DECIPHER database. The results showed that 71% of the somatic cancer SVs in COSMIC database are larger than 10 kb and all deletions/duplications that cause the CNV syndromes are larger than 10 kb.

***Reviewer Comments:** I would also recommend using the gold standard data set from GIAB HG002 which also includes high quality 10x genomics data.*

Response: Thank you for the comments. The benchmarking on GIAB HG002 genome was shown in Supplemental Figure 14. The GIAB SV calls set only contains insertions and deletions. Since most tools including Longranger, GROC-SVS, and NAIBR cannot detect insertions, we only benchmarked the performance on deletion detection.

We didn't find a good SV call set for benchmarking the other SV types on real data. Two recent studies^{1,2} reported SV calls of three family trios based on high-coverage long read sequencing data. We downloaded the two SV call sets from the dbVar database using the accession nstd162 and nstd152 and extracted the SV calls of GRCh38 coordinates. We found that the duplication calls reported by the two studies are largely inconsistent. For example, the nstd162 study contains 464 duplication calls of the NA19240 genome and the nstd152 study contains 1073 duplication calls of the NA19240 genome. Only 42 duplication calls are shared between the two studies. This is partly because the duplications are enriched in common repeats and segmental duplication regions. To further address this point, according to review 3's suggestion, we did additional simulation over known disease casual SVs (page 11-12, lines 198-205).

Responses to reviewer 3

The authors present LinkedSV, a new method for detecting structural variants in 10X data. This method uses the similarity between the sets of bar codes in different regions to help identify variants. The paper is well written, the process is interesting, and the performance of LinkedSV was better than other methods. I also liked the section about detection in WES, which is an area that needs better SV detection methods.

Response: Thank you for the nice summary of the manuscript and the tool. Our point by point responses are given below.

Reviewer Comments: *My major issue with is paper is its reliance on simulated data. While I am very sympathetic to the motivations of using simulations, especially considering that there are no good truth sets for SVs, it is difficult to validate the simulation itself. The authors biased their simulations toward repetitive regions based on the observation that these regions are more likely to harbor SVs. On the surface this seems like a good idea, but how faithful is the simulation to this biologic observation? To me, it seems more reasonable to simulation 10X over known SVs in these repetitive regions, instead of randomly picking something.*

Response: Thank you for the comments. To simulate over known diseases casual SVs, we downloaded a list of expert-curated deletions and duplications that are known to cause CNV syndromes involved in developmental disorders. This list was downloaded from the DECIPHER database, and it contained 67 CNV syndromes. Based on the deletions and duplications in this list, we simulated both germline and somatic WGS data sets and benchmarked the performances of LinkedSV as well as 5 other SV callers (Supplemental Figure 16). (These SVs are not cancer SVs, but somatic SVs are not only found in cancers, but also in other tissues, such as brain^{3,4}) The results were similar to those of our original simulations. According to reviewer 2's suggestion, we also added an additional benchmark on one real data set (the GIAB HG002 genome).

Reviewer Comments: *I am also troubled by the differences between the germline and the cancer simulation. In particular, that GROC-SVs went from matching LinkedSV's performance in one simulation to detecting nothing in the next, while the NAIBR went from being one of the worst methods to the only other method to detect anything. I understand that the allele balance shifts between these two, but detecting nothing makes me think something is wrong with the simulation or the way the tools were run. Other SV methods (not 10X methods) have done similar experiments, and I have never seen a report of a high-quality caller that "almost completely failed."*

Response: Thank you for the pointing this out. Here are the responses to your comments one by one.

1) *NAIBR went from being one of the worst methods to the only other method to detect anything.*

The main issue with NAIBR on the simulated germline data set was its low precision. After we examine our commands and the SV calls generated by NAIBR, we found that the benchmarking of NAIBR on the simulated germline data set was based on an early version of NAIBR, which had a bug that output small SVs of less than 1 kb, all of which were false positives. The benchmarking on the somatic data set was performed a few month later and used a later version of NAIBR, which did not have this bug.

To avoid the effect of software versions, and make sure our results are up-to-date, we performed the benchmarking studies again on the simulated WGS data sets using the latest released versions of all SV callers. The results are shown in the updated Figure 2a, 3a and 3b. The benchmarking on the WES data set also used the latest versions.

2) *In particular, that GROC-SVs went from matching LinkedSV's performance in one simulation to detecting nothing in the next. I understand that the allele balance shifts between these two, but detecting nothing makes me think something is wrong with the simulation or the way the tools were run.*

According to reviewer 2's suggestions, we also evaluated the performance of Delly and Lumpy on the simulated data sets (Figures 2a, 3a and 3b). The results showed that when the VAF is 10%, the recall rates of Delly and Lumpy were 27.7% and 7.2%, which are higher than GROC-SVs. In addition, the updated version of Longranger is able to detect 17.1% of the SVs from this data set. This indicates that the simulation is correct and some of the SVs can be detected even using the paired-end signals.

GROC-SVs is a germline SV caller. Therefore, its underlying statistical model may be different from a somatic SV caller, and may not be optimal for finding somatic SVs.

Reviewer Comments: *In the section that compared long-reads to linked-reads, I would like the authors to dig in a bit more as to why the SVs that were not detected using long reads but picked up by my LinkedSV are not false positives (beyond visual inspection). For example, the chr19 duplication spans a region that is densely tiled with microsatellites, which are a well known source of false positives for short read alignment. Maybe the reason that long reads do not show extra coverage here is that the long reads are correctly mapped whereas the short reads are not.*

Response: Thank you for the comments. We performed additional analysis of the two SV calls that were not detected by long reads as described below.

For the chr2 deletion, we were able to find the clipped reads at the breakpoints (Figure 7b-c). Further analysis of the clipped reads showed that this was a complex SV event. The 45 kb region in chr2 was deleted and a 6 kb sequence from the hs38d1 decoy sequence was inserted. As described in the manuscript (pages 19-20, lines 343-356), we aligned all the PacBio reads to a new reference genome with all sequences of GRCh38 plus hs38d1 and the sequence of the proposed variant allele. There were 33 reads spanning the chr2-hs38d1 junction, 48 reads spanning the hs38d1-chr2 junction and 13 reads spanning both junctions. *De novo* assembly of the reads aligned to the proposed variant allele generated a single contig of 42.7 kb, which also spanned both junctions (Supplemental Figure 9b). These analysis showed that the large deletion

event detected from both platforms and with PacBio long reads the details of complex SV events could be resolved.

For the chr19 duplication, we plotted the sequencing coverage near the duplication region (Supplemental Figure 10). The PacBio long reads also have extra coverage in this region.

Reviewer Comments: *I am sure that LinkedSV does better than other SV callers that do not consider the bar codes, but it would be interesting to see if there is any reduction in sensitivity when you start by looking for the barcode pattern.*

Response: Thank you for the comments. In our benchmarking studies (Figure 2a, Figure 3a and b, Figure 4a, Supplemental Figure 14, and Supplemental Figure 16), LinkedSV did not show a reduction in sensitivity for detection of large SVs, as the recall of LinkedSV is constantly higher than Delly and Lumpy. This is because the barcode information helps detected SVs when the paired-end signals were weak. However, the barcode information provides little help to resolve small SVs that are < 10 kb. To address these concerns, we have substantially improved LinkedSV, so that it now incorporates read depth and paired-end signals, and it now uses a local assembly-based method for very small SVs to achieve single-base resolution. By including these additional sources of information to the barcode information, the quality and accuracy of the tool are greatly improved as shown in the revised manuscript on both simulation and real data sets.

- 1 Audano, P. A. *et al.* Characterizing the Major Structural Variant Alleles of the Human Genome. *Cell* **176**, 663-675 e619, doi:10.1016/j.cell.2018.12.019 (2019).
- 2 Chaisson, M. J. P. *et al.* Multi-platform discovery of haplotype-resolved structural variation in human genomes. *Nat Commun* **10**, 1784, doi:10.1038/s41467-018-08148-z (2019).
- 3 Bedrosian, T. A., Quayle, C., Novaresi, N. & Gage, F. H. Early life experience drives structural variation of neural genomes in mice. *Science* **359**, 1395-1399, doi:10.1126/science.aah3378 (2018).
- 4 Nishioka, M., Bundo, M., Iwamoto, K. & Kato, T. Somatic mutations in the human brain: implications for psychiatric research. *Mol Psychiatry* **24**, 839-856, doi:10.1038/s41380-018-0129-y (2019).

REVIEWERS' COMMENTS:

Reviewer #1 (Remarks to the Author):

All my comments are addressed.

Reviewer #2 (Remarks to the Author):

The authors have addressed all my questions and concerns. It is nice to see that they improved their method to detect also smaller deletions and cleared up some formulations.

Just one minor comment to their response, but not relevant to the manuscript:

1. About the larger SV size impacting. The reason I am arguing against it is that we dont know if the larger CNV in the data based are more prevalent only because they are easier to detect. Especially since CNV caller have a limited size resolution compared to the SV callers.

Reviewer #3 (Remarks to the Author):

The authors address all of the issues I raised and have made the paper much paper. Nice work.

REVIEWERS' COMMENTS

Reviewers' comments

Reviewer #1 (Remarks to the Author):

All my comments are addressed.

Response: Thank you very much for evaluating our manuscript!

Reviewer #2 (Remarks to the Author):

The authors have addressed all my questions and concerns. It is nice to see that they improved their method to detect also smaller deletions and cleared up some formulations.

Just one minor comment to their response, but not relevant to the manuscript:

1. About the larger SV size impacting. The reason I am arguing against it is that we dont know if the larger CNV in the data based are more prevalent only because they are easier to detect. Especially since CNV caller have a limited size resolution compared to the SV callers.

Response: Thank you very much for evaluating our manuscript! We agree that large CNVs are easier to detect and added one sentence in Supplementary Note 2 so that readers are aware of this potential bias (page 42 of the Supplementary Information file, last sentence).

Reviewer #3 (Remarks to the Author):

The authors address all of the issues I raised and have made the paper much paper. Nice work.

Response: Thank you very much for evaluating our manuscript!